# On the importance of the albedo parameterization for the mass balance of the Greenland ice sheet in EC-Earth

Michiel M. Helsen[1], Roderik S.W. van de Wal[1], Thomas J. Reerink[1], Richard Bintanja[2], Marianne S. Madsen[3], Shuting Yang[3], Qiang Li[4], and Qiong Zhang[4]

[1]Institute for Marine and Atmospheric Research Utrecht, Utrecht University, The Netherlands
[2]Royal Netherlands Meteorological Institute, De Bilt, The Netherlands
[3]Danish Meteorological Institute, Copenhagen, Denmark
[4]Bolin Centre for Climate Research, Department of Physical Geography, Stockholm University, Stockholm, Sweden

*Correspondence to:* Michiel Helsen (m.m.helsen@uu.nl)

**Abstract.** The albedo of the surface of ice sheets changes as a function of time, due to the effects of deposition of new snow, ageing of dry snow, bare ice exposure, melting and runoff. Currently, the calculation of the albedo of ice sheets is highly parameterized within the earth system model EC-Earth, by taking a constant value for areas with thick perennial snow cover. This is an important reason why the surface mass balance (SMB) of the Greenland ice sheet (GrIS) is poorly resolved in the model. The purpose of this study is to improve the SMB forcing of the GrIS, by evaluating different parameter settings within a snow albedo scheme. By allowing ice sheet albedo to vary as a function of wet and dry conditions, the spatial distribution of albedo and melt rate improves. Nevertheless, the spatial distribution of SMB in EC-Earth is not significantly improved. As a reason for this, we identify omissions in the current snow albedo scheme, such as separate treatment of snow and ice and the effect of refreezing. The resulting SMB is downscaled from the lower resolution global climate model topography to the higher resolution ice sheet topography of the GrIS, such that the influence of these different SMB climatologies on the long-term evolution of the GrIS is tested by ice sheet model simulations. From these ice sheet simulations we conclude that an albedo scheme with a short response time of decaying albedo during wet conditions performs best with respect to long-term simulated ice sheet volume. This results in an optimised albedo parameterization that can be used in future EC-Earth simulations with an interactive ice sheet component.

## 1 Introduction

Ice mass loss from the Greenland ice sheet (GrIS) is currently an important and accelerating contributor to sea level rise (Shepherd et al., 2012; Vaughan et al., 2013; Velicogna et al., 2014; Kjeldsen et al., 2015). Ice sheet models are used in order to understand ice sheet evolution and to generate projections for the future. The outcome of ice-sheet model (ISM) simulations of the GrIS is strongly constrained by the surface mass balance (SMB) forcing, i.e., the net result of mass gain at the surface of the ice sheet by snowfall and rain, redistribution of snow through wind action, and mass loss by sublimation and meltwater runoff. One can use different sources to obtain a SMB forcing for ISM simulations, ranging from observations, regional climate model

(RCM) output and global climate model (GCM) output. This choice is usually determined by the nature of the ISM simulation: future projections rely on climate model output, where observations of snow accumulation can be used for present-day analysis.

For centennial-scale present-day and near-future ice sheet simulations, high-resolution SMB products from RCMs such as MAR (Fettweis et al., 2013) or RACMO2 (Van Angelen et al., 2014) are widely used to drive GrIS model simulations (e.g. Lea et al., 2014; Colgan et al., 2015). Climate models that include a multi-layer snow model are increasingly able to simulate relevant processes as snow densification, percolation and refreezing, which improves the physics of the simulated near-surface climate and firn processes. Moreover, high-resolution regional climate modelling is generally regarded as superior to observation-based SMB products, due to the better spatial coverage of the models, which offer more details than interpolating inhomogeneous available observations.

However, the performance of RCMs is to a large extent determined by the quality of the forcing at the lateral boundaries and the sea surface. For the present-day configuration this is not a problem due to the existence of good quality global reanalysis data (e.g. Uppala et al., 2005; Dee et al., 2011). Nevertheless, this is not possible for paleo-studies nor for future projections, hence RCMs then have to rely on changing lateral boundary conditions from simulations of GCMs (e.g. Hostetler et al., 2000; Van de Berg et al., 2011; Maris et al., 2014). Consequently, a more direct approach for those cases would be to calculate the SMB directly within the GCM. However, this requires a high resolution GCM, or alternatively downscaling techniques. In addition, millennial-scale (and longer) ice sheet simulations lead to significant changes in ice sheet surface elevation and extent. To account for feedbacks from the growth or decay of ice sheets on the climate system itself, an interactive ISM component needs to be included in a GCM.

Different GCMs with interactive ISM components exist, all with their own downscaling solutions to retrieve a meaningful SMB forcing over ice sheets. Lipscomb et al. (2013) compute SMB by solving the surface energy balance in combination with a multilayer snow model. Moreover, to overcome differences in resolution between the Community Earth System Model (CESM) and the ISM grid, they apply the multilayer snow model in 10 elevation classes for each grid cell within the land surface model. For each elevation class, prognostic calculations of albedo, melt, refreezing and runoff are carried out, driven by downscaled turbulent fluxes (using Monin-Obukhov similarity theory), atmospheric surface temperature (using a globally uniform lapse rate of $6\,\mathrm{K\,km^{-1}}$) and specific humidity (assuming constant relative humidity), from the mean grid cell elevation to the elevation of each class. Incoming radiative fluxes and precipitation are not downscaled. A more simplified approach is the use of a positive degree day (PDD) model to estimate surface melt from near-surface temperatures, which is estimated using a lapse rate. This method is applied by e.g. Ziemen et al. (2014) to force the Parallel Ice Sheet Model (PISM) with the ECHAM5 GCM. A review of the status and challenges related to earth system models with interactive ice sheets is given in Vizcaíno (2014).

SMB calculations using an energy balance approach strongly rely on the parameterization of albedo. Since snow albedo is high, only a small portion of incoming solar radiation is absorbed by the snow, implying a strong sensitivity of SMB to the albedo values. Local melting leads to rapid snow metamorphism and a related drop of the albedo (e.g. Lefebre et al., 2003), increasing absorbed solar radiation, and thereby providing more energy that can contribute to melting of the snow. As such, even a small perturbation of the albedo can amplify changes in the radiation budget of the snow pack. In the absence of a snow

layer, albedo of bare ice is much lower, and depends on impurity content (Greuell and Genthon, 2004; Bøggild et al., 2010; Wientjes and Oerlemans, 2010; Wientjes et al., 2011). Albedo of dry snow decreases more slowly, as a function of changes in snow grain size and shape, but is also influenced by impurity content, solar zenith angle and cloudiness (Warren, 1982; Greuell and Genthon, 2004; Gardner and Sharp, 2010). As a consequence albedo schemes need to be treated carefully. Each climate

model has its own albedo parameterization, ranging from the most simple solution of a prescribed constant albedo over ice sheets (e.g. HadGEM2, ECHAM5; Martin et al., 2011; Ziemen et al., 2014), snow albedo as a diagnostic variable depending on the time after snow deposition (e.g. CNRM, Douville et al., 1995; Voldoire et al., 2013), to albedo as a prognostic variable based on radiative transfer computations, effective snow grain size, solar zenith angle, and presence of carbon or dust (e.g. CESM, Lipscomb et al., 2013).

To better understand changes in the mass of the ice sheet and in its interaction with the climate system for paleo-studies, present-day simulations and future climate projections, we aim to include an ISM component in the earth system model EC-Earth (Hazeleger et al., 2010). As a step in this direction, the quality of the SMB forcing from EC-Earth over the GrIS should be improved. In this study, we investigate how the snow albedo parameterization over ice sheets influences the modelled SMB estimates. The original albedo parameterization in EC-Earth is adjusted to accommodate a more realistic seasonal evolution of

the albedo, which leads to a more realistic calculation of snow melt and hence also the SMB. The SMB over GrIS resulting from simulations using various parameters in the albedo parameterization scheme are compared and evaluated using the SMB climatology estimated with the high resolution RCM RACMO2. Subsequently, we compare further the downscaled SMB on the higher-resolution ice sheet topography, using spatial relations between SMB and surface elevation (SMB gradients method, Helsen et al., 2012). As a final step, we carry out ISM simulations to simulate the steady-state size of the GrIS as a function of

the different SMB climatologies. In this way, we use the albedo parameterization as a calibration tool, such that we determine the optimal combination of parameters in the albedo parameterization, that results in a SMB forcing which in turn leads to a steady-state GrIS simulation close to the present-day size. Our experiments also identify which modifications are necessary for further improvements of ice sheet mass balance within GCMs. Such improvements regarding the description of ice sheets in GCMs is vital for a better understanding of changes in ice sheets in the past, present and future.

**2   Albedo parameterization**

Ice sheet SMB is the net result of accumulation of snow, and ice mass loss by sublimation, surface melt and subsequent run-off. The amount of melt is a function of the surface energy balance (SEB) components, which vary widely in space and time over the ice sheet. An important source of melt energy is delivered by shortwave radiation, but due to the high albedo of snow, most of the incoming shortwave radiation is reflected back into space. Therefore, the snow albedo evolution exerts a strong control

on the ice sheet SEB, and thus on the SMB.

In cold, dry conditions, snow albedo over ice sheets remains relatively high (Loth et al., 1993), although with time snow grains grow, which effectively reduces the albedo (Warren, 1982; Gardner and Sharp, 2010). However, this process is generally slow at low temperatures. Observed broadband albedo values in the accumulation area of the GrIS are in the range 0.75–0.83

(Alexander et al., 2014). The presence of liquid water has a darkening effect on snow: water in snow strongly accelerates grain growth and fills in voids between the snow grains. Both of these effects increase the effective grain size, thus rapidly decreasing the broadband albedo of wet snow (e.g. Gardner and Sharp, 2010). Liquid water ponding on the glacier surface, removal of the firn layer and subsequent melting of the bare ice are associated with broadband albedo values of $\sim 0.5$ (Van de Wal and Oerlemans, 1994; Van de Wal et al., 2005; Van den Broeke et al., 2008, 2011; Alexander et al., 2014). The accumulation of impurities, microorganisms, dust or debris on the glacier surface can further decrease albedo to values $< 0.3$ (Greuell and Genthon, 2004; Bøggild et al., 2010; Wientjes and Oerlemans, 2010; Wientjes et al., 2011). To minimise the bias introduced by uncertainty in the albedo, it can be a solution to use observed albedo (e.g. MODIS, Box et al., 2017), but this is only a solution for experiments describing the recent observational period, and not possible for paleo and future simulations.

## 2.1 Albedo in EC-Earth Standard Set-up

EC-Earth (version 2.3) participated in CMIP5 (Taylor et al., 2012). The model was evaluated against observations, reanalysis data and other coupled atmosphere-ocean-sea ice models, and its performance was good, in terms of the mean state, spatial patterns, seasonal cycle and variability of present-day climate (Hazeleger et al., 2010, 2012). We use the updated EC-Earth version 3.1, i.e., ECMWF's Integrated Forecasting System (IFS) on a spectral resolution truncated at wave number 255 (T255), with 91 vertical atmospheric levels. Snow albedo ($\alpha_{sn}$) is calculated in EC-Earth as part of the Hydrology Tiled ECMWF Scheme of Surface Exchanges over Land (HTESSEL; Viterbo and Beljaars, 1995) and it includes an explicit snow scheme, which consists of one layer of snow, with a time-evolving density (Dutra et al., 2010). Heat fluxes are calculated through the snow layer, and snow liquid water capacity is approximated as a function of density and snow mass. Albedo varies as a function of time; linearly decreasing for dry snow and exponentially decreasing for melting conditions. When the flux of fresh snow exceeds a certain threshold, $\alpha_{sn}$ is reset to its maximum value ($\alpha_{max}$=0.85). However, $\alpha_{sn}$ is assigned a fixed value ($\alpha_{per}$=0.80) for areas where the snow layer has reached the maximum value of 10 m. These are typical conditions that prevail over ice sheets, and as such snow albedo is in effect constant over Greenland in the standard set-up of EC-Earth:

$$\alpha_{sn} = 0.80 \tag{1}$$

A continuous growth of the snow layer due to a positive SMB over perennial snow covered areas is not desirable in climate models, because it violates water conservation. Therefore, the snow layer thickness is set to a maximum (10 m w.e.) in EC-Earth. When snowfall occurs on top of a snow layer with maximum thickness, the excess snow is returned to the hydrological cycle as runoff, i.e. by adding it to the meltwater flux. This aspect of the treatment of ice sheets has been left unchanged, but to enable a time-varying snow albedo, the snow model has been adjusted as described below. Melting of snow is based on the surface energy balance and only occurs when the energy flux to the surface is positive.

## 2.2 Adjustments for Ice Sheets

Efforts have been undertaken by groups in Stockholm (Bolin Center), Copenhagen (DMI) and Utrecht (IMAU-KNMI), to include an interactive ice sheet component within EC-Earth (Svendsen et al., 2015). As part of this effort, a new surface type

"ice sheet" is introduced in the land surface module HTESSEL of the Integrated Forecasting System (IFS) atmosphere model within EC-Earth. GrIS areal extent from Howat et al. (2014) is mapped to the T255 resolution using the mapping method OBLIMAP (Reerink et al., 2010, 2016), and is read by EC-Earth in the initialisation phase. All runs described in this study use the same GrIS domain.

Snow albedo can be prescribed differently for this ice sheet surface type. However, the land surface model does not distinguish between snow or ice, nor is snow grain size simulated. Therefore, the basis of the snow scheme applied over ice sheets is still from the original scheme for seasonal snow by Dutra et al. (2010), but each of the groups mentioned above has made different choices for the parameter settings. Here, these different choices for snow albedo are briefly described.

Instead of using a constant value of $\alpha_{sn}$=0.80, a time-evolving scheme is adopted, using different minimum and maximum
values. When fresh snow accumulates the snow albedo is reset to a maximum value ($\alpha_{max}$). This is done similarly by all groups:

$$\alpha_{sn}^{t+1} = \alpha_{sn}^t + min\left(1, \frac{F\Delta t}{10}\right)\left(\alpha_{max} - \alpha_{sn}^t\right) \tag{2}$$

$\Delta$t represents the time step in the model (in hours). When the snowfall rate (F) exceeds 10 kg m$^{-2}$ h$^{-1}$, snow albedo is fixed at $\alpha_{max}$. A continuous reset for small amounts of snowfall is implemented to reduce the importance of small amounts of
snowfall on albedo. However, $\alpha_{max}$ depends on the scheme which is used (Table 1).

When the snow layer is not refreshed by new snowfall, albedo can decrease with time. A distinction is made between the slow effect of ageing of dry snow (slow decay of albedo) and the faster effect of wet, melting snow. So-called "wet" conditions are recognised when either liquid water is present (due to melting or rainfall), or when snow temperature is within 2K of the melting point.

For dry conditions, a slow exponential decay is applied toward a minimum firn albedo value ($\alpha_{firn}$=0.75), parameterizing the effect of grain growth on snow albedo:

$$\alpha_{sn}^{t+1} = \left(\alpha_{sn}^t - \alpha_{firn}\right)e^{\left(-\frac{\Delta t}{\tau_{firn}}\right)} + \alpha_{firn} \tag{3}$$

We use an e-folding time scale $\tau_{firn}$ of 30 days. Parameterizing the effect of dry metamorphism of snow by applying an exponential function has also been done by e.g. Bougamont et al. (2005), who also used $\tau_{firn}$=30 days.

For wet conditions, a faster exponential decay of albedo can be prescribed towards a minimum value ($\alpha_{min}$), with an e-folding time scale ($\tau_{melt}$):

$$\alpha_{sn}^{t+1} = \left(\alpha_{sn}^t - \alpha_{min}\right)e^{\left(-\frac{\Delta t}{\tau_{melt}}\right)} + \alpha_{min} \tag{4}$$

Note that we do not make an explicit distinction between albedo of snow and ice. When all snow is melted on a grid point within the ice sheet mask, we use a constant minimum albedo ($\alpha_{min}$). The parameter values as used by different groups are
summarised in Table 1.

### 2.2.1 Albedo Scheme Stockholm (Sto)

In the "Sto" albedo scheme, fresh snowfall is given the value $\alpha_{max}$=0.80. The effect of ageing of dry snow on the albedo is neglected, hence no decrease of $\alpha_{sn}$ is applied for dry conditions. Instead of applying equation 3, $\alpha_{sn}$ is kept constant during dry conditions:

$$\alpha_{sn}^{t+1} = \alpha_{sn}^{t} \tag{5}$$

For wet conditions, the "Sto" albedo scheme uses the exponential decay (equation 4) with an e-folding time scale ($\tau_{melt}$) of 4 days (Table 1). This time scale is similar to the original snow albedo scheme for seasonal snow by Douville et al. (1995) and Dutra et al. (2010).

### 2.2.2 Albedo Scheme Copenhagen (Cph)

The albedo scheme "Cph" assigns a higher value to fresh snow compared to the "Sto" albedo scheme: $\alpha_{max}$=0.85. For dry conditions, scheme "Cph" applies the slow exponential decay (equation 3), parameterizing the effect of grain growth on snow albedo.

In contrast to the "Sto" scheme, the "Cph" scheme prescribes an instantaneous response of the albedo in wet conditions:

$$\alpha_{sn}^{t+1} = \alpha_{min} \tag{6}$$

The rationale behind this choice is that by infilling the voids between the snow grains, the presence of water within the snow can have an instantaneous effect by increasing the effective grain radius. However, it should be noted that this immediate effect is generally regarded as weak, and that the dominant effect of the presence of water is a strong acceleration of grain growth (Gardner and Sharp, 2010).

In case the conditions switch from wet to dry, the albedo is reset to $\alpha_{refr}$=0.65, to account for the relatively low albedo of
refrozen meltwater.

### 2.2.3 Albedo Scheme Utrecht (Utr)

For dry conditions, the "Utr" scheme adopts the same choices as the "Cph" scheme (equation 3). A set of simulations is performed in which the value of $\alpha_{max}$ is varied from 0.80 to 0.85.

For wet conditions, we test different e-folding time scales for the exponential decay (equation 4), varying from 0 to 4 days, to
account for the influence of rapid snow metamorphism on alb edo during wet conditions. We also perform one experiment with an extremely low albedo of 0.45 that is attained instantaneously when wet conditions prevail. This values is consistent with bare ice. Next to that, we use somewhat lower minimum albedo values for melting snow and snow with refrozen meltwater (Table 1), which are more in line with observations in the ablation area of the GrIS (Van de Wal and Oerlemans, 1994; Van den Broeke et al., 2008, 2011; Alexander et al., 2014).

## 2.3 Performance of Albedo Schemes

To test the performance of the different albedo schemes, we performed atmosphere-only runs using EC-Earth version 3.1. Simulations start on 01-01-1990, and are forced with 1990-2012 sea surface temperatures from the ERA-Interim reanalysis. Figure 1 illustrates the temporal evolution of $\alpha_{sn}$ in the different simulations at four locations over the GrIS: the western ablation area, the northeastern ablation area, Summit and the southeastern high accumulation area. Locations are shown in Figure 2. For clarity, only the first simulated year (1990) is shown in Figure 1, which allows identification of the influence of individual snowfall events. Hence, initial values of $\alpha_{sn}$ are identical for all the simulations except for the reference run, which has a constant value of $\alpha_{sn}$=0.80 throughout the whole simulation. Note that qualitatively, a similar picture arises for other years (not shown). Since the simulated climate is freely evolving after initialisation, a direct comparison with observed time series is not meaningful.

The albedo time series of the different schemes have some common aspects: albedo fluctuations are minor during the dry wintertime conditions, whereas albedo drops rapidly to lower values during the ablation season in all runs. The "Sto" scheme (red line) shows the least variations, due to its characteristic of a constant albedo during cold, dry conditions.

The "Cph" scheme allows more variability in $\alpha_{sn}$ during dry conditions: the exponential decay of ageing dry snow is especially clearly visible in Figure 1b, which is representative for the cold and dry interior. During wet conditions, $\alpha_{sn}$ instantly drops to $\alpha_{min}$, which can be seen in Figure 1a, a relatively dry site in northeast Greenland. The interplay of episodes of melt and accumulation of fresh snow introduces a large variability, which is especially apparent in Figure 1d.

In all figures, we only show results of one of the five versions of the Utrecht scheme: "Utr-8", which uses $\tau_{melt}$=1 day. This scheme performs best in terms of the resulting climate forcing and steady-state ISM simulations (see below). It results in more rapid and larger transitions between dry and wet conditions than the "Sto" and "Cph" schemes (Figure 1c and d). In conditions when ablation prevails, $\alpha_{sn}$ is lower, since we chose lower values for $\alpha_{min}$ and $\alpha_{refr}$, which are more in line with measured albedo over ice sheets (Van de Wal and Oerlemans, 1994; Van den Broeke et al., 2008, 2011; Alexander et al., 2014). It should be noted that timing of snowfall events can differ between the simulations. This is a consequence of the different albedo schemes: temperatures are slightly different and the atmospheric circulation changes.

The different albedo parameterizations give rise to subtle differences in the pattern of average albedo over Greenland, because albedo is more or less constant during winter and varies in summer depending on the melt. Results for the summer season (23-yr mean JJA albedo values) are shown in Figure 2. Observations indicate that Greenland JJA albedo should be within the range 0.75-0.85 in the accumulation area, and 0.5-0.6 in the ablation area (Van de Wal and Oerlemans, 1994; Van den Broeke et al., 2008, 2011; Alexander et al., 2014). Despite the assumption that $\alpha_{sn}$ does not decrease in dry conditions, the "Sto" albedo scheme shows lower JJA albedo in the accumulation area than the other schemes. This is caused by the lower value for $\alpha_{max}$: the "Sto" scheme uses 0.80, whereas "Cph" and "Utr-8" schemes use 0.85. With respect to the observed JJA albedo in the ablation area, the "Sto" albedo pattern (Figure 2a) has somewhat too high albedo. The "Cph" scheme leads to a more pronounced difference between the accumulation area and the ablation area (Figure 2b). The decrease of the albedo towards the ice sheet margin can be attributed to the choice of a constant value for $\alpha_{min}$ for melt conditions. The "Utr-8"

albedo scheme leads to an even more pronounced difference between accumulation and ablation area (Figure 2c), with values more in line with observed values (Van de Wal and Oerlemans, 1994; Van den Broeke et al., 2008, 2011; Alexander et al., 2014), mainly due to the choice for a lower value of $\alpha_{min}$ compared to the other schemes. The rapid exponential decay in scheme "Utr-8" ($\tau_{melt}$= 1 day) serves as a good compromise between the instantaneous effect of melt ("Cph") and the delayed response of 4 days ("Sto").

For a spatial comparison, we compare our simulated JJA albedo fields with albedo from the high-resolution regional atmospheric climate model RACMO2 (1960-1989) (Van Angelen et al., 2014). Forced with ERA-40 (Uppala et al., 2005) and ERA-interim (Simmons et al., 2007) at its lateral boundaries, RACMO2 is run on an 11 km horizontal resolution, and it has 40 sigma levels in the vertical. Snow albedo in RACMO2 is based on effective snow grain size in a more sophisticated snow scheme (Kuipers Munneke et al., 2011). RACMO2 uses observed (MODIS) albedo in case the snow layer is absent, so for these regions we effectively compare with satellite-observed albedo. To facilitate a comparison, EC-Earth fields and RACMO2 fields are mapped onto a 20 km rectangular grid (also used for the ISM simulations, see below). The difference with JJA albedo from RACMO2 is plotted in Figure 2d-f. This comparison reveals that the different schemes in EC-Earth slightly underestimate albedo in the upper accumulation area, but biases are small ($<0.05$). The bias becomes larger in a narrow zone in the lower accumulation area, where EC-Earth JJA albedo is lower than RACMO2 albedo. This is a firn-covered area of the ice sheet, where our snow albedo scheme falls short, as it does not distinguish between snow and ice. Hence melt conditions lead to (too) fast declining albedo values, leading to slightly deteriorated RSMD values. Instead, snow albedo in RACMO2 remains higher for melt conditions in this area. However, in the lower ablation zone the bias becomes positive, and these biases are smallest for the Utr-8 scheme. This positive bias in the lower ablation zone indicates that in some locations, summertime albedo can drop to values lower than 0.5 (minimum snow albedo used in this study).

## 3 Simulated Surface Mass Balance

Differences in albedo affect the SMB through the SEB. Here, we calculate SMB using an energy balance approach. Model simulated six-hourly accumulated values of the net shortwave radiation ($SW_{net}$), longwave radiation ($LW_{net}$), the turbulent fluxes of sensible heat (SH) and latent heat (LH) are used to calculate the SEB over the GrIS domain:

$$SEB = SW_{net} + LW_{net} + SH + LH \tag{7}$$

In addition, skin temperature ($T_s$) of the ice sheet surface is inferred from upwelling longwave radiation, using the Stefan-Boltzmann law, assuming an emissivity of 1. For each 6-hourly interval with a positive SEB and $T_s$ within 1 K of the melting point it is assumed that all available energy is used to melt snow. SMB is consequently calculated as the sum of total solid precipitation, evaporation and melt. As such, we assume that all melt immediately runs off, hence ignoring refreezing of percolating meltwater.

Uncertainties exist in estimates of SMB. Contemporary best estimates of SMB integrated over the GrIS are based on RCMs and range between 388 $\pm$103 Gt yr$^{-1}$ (1980–1999, Fettweis et al., 2013) and 406 $\pm$98 Gt yr$^{-1}$ (1960–1990, Van Angelen

et al., 2014). These time periods are still relatively short in climatological terms for an ice sheet. SMB estimates covering a much longer time period also exist, but typically rely on a PDD model to estimate melt and runoff, and combine data from reanalysis, weather stations and ice cores to calculate a SMB record. Following this approach, Hanna et al. (2011) estimated a GrIS SMB value of $368 \pm 129$ Gt yr$^{-1}$ for the period 1871–2010, whereas Box (2013) found an average SMB value of 459

$\pm 100$ Gt yr$^{-1}$ for the period 1840–2010.

Here we compare different components of simulated SMB fields (1990-2012 averages), resulting from the use of different albedo schemes in EC-Earth, with components of the SMB climatology from RACMO2 (1960-1989), which has been extensively compared with observed SMB data (Ettema et al., 2009; Van Angelen et al., 2014).

The patterns of mean annual snowfall are very similar in all runs, therefore we only show the 23-yr (1990-2012) snowfall
climatology from the reference run with the unchanged albedo scheme in Figure 3a. A comparison with RACMO2 (Van Angelen et al., 2014, Figure 3b and c) shows that the general pattern of snowfall is well-captured in EC-Earth, with accumulation maximums in the southeast, a dry central and northern Greenland, and weak local maximums along the western ice margin. The differences can be attributed to the lower resolution of EC-Earth, causing a more widespread zone of maximum snowfall, which leads to a wide band of overestimated snowfall over the southern part of the GrIS (positive values in Figure 3c). The
lower resolution misses more pronounced local maximums in snowfall along the southeastern ice margin, which translates to local underestimation of snowfall (negative values in Figure 3c). This is in broad agreement with results from Franco et al. (2012), who did RCM experiments with MAR on a range of spatial resolutions. Ice-sheet integrated snowfall of the EC-Earth reference run is 95 Gt yr$^{-1}$ larger than snowfall from RACMO2 (Table 2), but uncertainty estimates based on the interannual variability from these fields are quite large.

The differences in albedo directly affect simulated melt (Figure 4). Melt is simulated in the southern part and on relatively low-lying areas on the GrIS. As expected, more melt is simulated with decreasing summertime albedo. In reality, the most pronounced (and extensive) ablation area of the GrIS is situated along the western margin at $\sim 67°$ N, but in EC-Earth this region does not stand out as a major ablation area. Compared to RACMO2, melt is underestimated for this area in all runs (Figure 4e-h), but overestimated in the northwestern, south and southeastern margin. RMSD values of melt indicate that the
"Utr-8" performs best with respect to reproducing the melt pattern.

The overestimation of melt may be due to a difference in topography of the GrIS as represented in EC-Earth and in reality, which will be investigated below. The differences in melt rate between EC-Earth and RACMO2 may also be influenced by biases in incoming radiation. However, a comparison of incoming shortwave and longwave radiation between EC-Earth and RACMO2 reveals that the differences in incoming shortwave radiation are mostly small over the ice sheet. Downwelling
longwave radiation is slightly higher in EC-Earth than in RACMO2, suggesting warmer air over the ice sheet in EC-Earth. This difference is most pronounced over the ice marginal area. From this we conclude that biases in incoming radiation do not play an important role in explaining the pattern of the difference in melt rate.

An additional effect of the use of a time-evolving albedo scheme is an increase of interannual variability in melt (Table 2). Qualitatively, this can be explained by the snow-albedo feedback: years with little snowfall will experience more melt due to a
low albedo, whereas in years with frequent snowfall albedo is kept high, and melt rates remain lower.

A comparison of the simulated SMB distribution (1990-2012 averages, Figure 5), resulting from the use of different albedo schemes in EC-Earth, with the SMB climatology from RAMCO2 reveals that the general distribution of SMB over Greenland is to a reasonable extent captured by EC-Earth: high accumulation in the southeast, a zone of high SMB over the western GrIS, a dry northern interior, ablation areas along the western, northwestern and northeastern margins. However, the ablation areas in EC-Earth are locally not large enough, and also the amount of ablation is underestimated. Note that the ice mask used in EC-Earth is somewhat larger than the ice mask in RACMO2, so the difference fields (Figure 4e-h and 5 e-h) are only shown for common grid points.

We can compute ice-sheet integrated values of the SMB components by summing over the entire ice sheet area (Table 2). All numbers for EC-Earth are computed on the same ice sheet mask (Howat et al., 2014), mapped to a 20 km rectangular grid; values outside this mask are disregarded. Note that ice-sheet integrated SMB values for EC-Earth can be computed from the components in Table 2 (SMB=SF-E-R). This is not the case for RACMO2, since this SMB calculation also includes a rain fraction, which is assumed to directly run-off in the calculation of SMB in EC-Earth.

Using the standard albedo scheme in EC-Earth, this results in a remarkable good match with ice-sheet integrated SMB from RACMO2, but in fact the spatial distribution of the mismatch shows that the standard albedo scheme in EC-Earth leads to a large underestimation of the major ablation areas, which is compensated by ablation areas in the south that are much larger than in RACMO2. Next to that, the larger accumulation in EC-Earth also influences the ice-sheet integrated SMB. The revised albedo schemes have led to a somewhat better representation of the ablation areas, indicating better agreement of the melt rates with respect to the RACMO2 estimate. However, whereas melt rates are simulated better, the match with ice-sheet integrated SMB values deteriorates with respect to RACMO2. This can be largely attributed to the neglect of refreezing in the SMB budget (Table 2).

This can also be illustrated by a closer inspection of the SMB pattern and its components at the western margin of the GrIS at $\sim 67°$ N. SMB is strongly overestimated in the reference run of EC-Earth in this area (Figure 5a and e). This is not surprising, considering the constant value of $\alpha_{sn}$=0.80 applied in this run, strongly limiting the role of absorbed solar radiation in SEB, leading to an underestimation of the melt rate. When $\alpha_{sn}$ is allowed to decrease in response to the occurrence of meltwater, the underestimation of the melt becomes smaller. The albedo scheme "Utr-8" even leads to lower SMB values compared to RACMO2 in the higher parts of the western ice sheet margin at $\sim 67°$ N (blue colours in Figure 5h), despite a local underestimation of the melt (Figure 4h), and only a minor difference in snowfall (Figure 3h). The key factor that explains this mismatch is our assumption that all melt translates into runoff, i.e. the buffering effect of refreezing is neglected.

Apart from the quite simple snow scheme, the relatively coarse resolution (a spectral resolution of T255 translates to $\sim 78$ km horizontal grid spacing) is an important factor that hampers a realistic representation of the SMB over Greenland. In some areas, steep topography cannot be resolved, producing associated errors in SMB. To downscale the SMB field from EC-Earth to a higher resolution topography that is used for ISM simulations, we use the SMB gradient approach (Helsen et al., 2012). Local spatial gradients of SMB as a function of surface elevation are used to correct the SMB for a mismatch between the surface topography in the climate model and the true ice sheet topography. For the latter we use the surface elevation field and ice sheet mask from Howat et al. (2014), regridded to a 20 km resolution using the mapping method OBLIMAP (Reerink

et al., 2010, 2016) to facilitate ISM simulations (see below). SMB gradients are computed for each SMB field, and resulting downscaled SMB fields are shown in Figure 6.

The overall effect of this procedure is an increase in melt and a decrease in SMB (Table 2). This can be mainly attributed to an increase of low-lying topography within the ablation zone of the ice sheet, resulting in more negative SMB. RMSD values of downscaled SMB are not improved, but regionally we can see an improved pattern of increased ablation area, which is of importance for ice sheet simulation (see below). This is particularly the case on the western ablation area at ∼67° N: it can be seen that SMB on the EC-Earth topography was not negative enough in the lower reaches in the ablation zone, whereas it was too low in the upper part of the ablation zone (Figure 5). Although this pattern of mismatch is still present, it is improved due to the downscaling step. The decrease in RACMO2 SMB after regridding is due to a larger ice sheet mask in the ISM, adding primarily ablation area.

## 4  Ice Sheet Simulations

Finally we test the effect of the different parameter values in the albedo parameterization on long-term simulations of a GrIS ISM. We performed simulations using the 3D thermomechanical ISM IMAU-ICE (previously known as ANICE, Van de Wal, 1999a, b; Bintanja and Van de Wal, 2008; Graversen et al., 2011; Helsen et al., 2012, 2013; De Boer et al., 2014), based on the Shallow Ice Approximation (SIA, Hutter, 1983). No ice shelf dynamics are included; as soon as the ice advances into ocean water and its thickness is not large enough to stay grounded, it breaks off. As such, calving by means of a floatation criterion is included, but calving physics are not explicitly incorporated.

The ISM calculates ice flow, thermodynamics, and bedrock response on a rectangular domain of 141x91 grid points with a grid spacing of 20 km. Initial bedrock elevation and ice thickness are taken from Bamber et al. (2013) and Howat et al. (2014). Initialisation of the internal ice temperatures is obtained by using the Robin solution based on surface temperature and SMB (Van Angelen et al., 2014) and a spatially distributed geothermal heat flux (Shapiro and Ritzwoller, 2004). Internal ice temperatures in the ablation zone are initialised as a linear profile between the surface temperature and the pressure melting point at the ice-sheet base.

Several 25-ky runs are carried out using the 23-year mean SMB forcing fields resulting from the different choices in albedo parameters. To take into account the influence of topography changes on surface temperature, a uniform lapse rate of -7.4 K km$^{-1}$ is applied. The influence of topography changes on SMB (height-mass balance effect) is parameterized by calculating a new SMB forcing field each time step using SMB gradients (Helsen et al., 2012). These gradients are computed for each grid point, from a linear regression between surface elevation and mean SMB values in an area with a radius of 150 km. This allows the ice sheet to advance outside the initial ice sheet mask, but also can lead to substantial retreat when ice sheet thinning occurs (Helsen et al., 2012).

After 25-ky of simulation, all but one of the simulated ice sheets are larger than the initial (present-day) volume and area, and are close to steady-state (Figure 7). Only the ice sheet forced by the "Utr-9" climatology resulted in ice sheet collapse. This is expected, as it is the only climatology with an initial negative SMB (Table 2), which suggests that the $\alpha_{min}$ value of 0.45

of albedo scheme "Utr-9" is too low. However, this result might be different if the model would account for refreezing, which would lead to a higher SMB, perhaps up to the point that the ice sheet will be stable. Hence, our results are strongly determined by the characteristics of our snow scheme.

The ISM run forced by the RACMO2 climatology results in an ice volume and area most comparable to the present-day state (Howat et al., 2014), in spite of the fact that several EC-Earth schemes result in lower ice-sheet integrated SMB (Table 2). This points out that the spatial distribution of SMB is important to the evolution of the ice sheet, rather than the overall numbers.

Figure 8 shows steady-state ice sheet elevation, and the difference of ice thickness with respect to the present-day. A common feature is the negative anomaly in ice thickness in the ice sheet interior, which is commonly seen in SIA type models, but can also be partly attributed to the fact that these simulations are in steady-state with the present-day climate, whereas the current GrIS contains colder ice originating from the last glacial period.

Another common feature in all simulations is ice growth (almost) all around the ice sheet. We cannot simply attribute this to an underestimation of the total melt, as it also occurs in the run forced with the RACMO2 climatology, and even in EC-Earth forced runs with more melt (and thus runoff) than the RAMCO2 climatology. Rather, the ice sheet thickening at the margins is a combination of different factors. Local ablation zones are not everywhere well captured by the SMB products, when the resolution of the climate model is not fine enough. In these cases, applying the downscaling method does not make much of a difference, as the resulting SMB gradients are not steep enough. This underestimation of ablation area results in ice sheet advance. Consequently, the height – mass balance effect results in an amplification of the initial perturbations. In addition, SIA-type ISM simulations also suffer from lack of fast-flowing outlet glacier dynamics, which explains part of the thickening in all areas where the ice sheet predominantly loses its mass through iceberg calving of narrow outlet glaciers. This is also caused by the limited spatial resolution of 20km which does not resolve most of the fjord systems, which hampers a correct simulation of ice loss due to calving of outlet glaciers. The ice sheet thus needs to advance to a point in contact with the ocean in order to lose its mass by calving.

From all ice sheet simulations forced with EC-Earth climatologies, we regard the simulation "Utr-8" as the best match with the observed ice sheet thickness and extent. It has the lowest RMSD value (Figure 8f-j), and the simulated ice volume and area are closest to the present-day observed value. The limited mismatch of the southwestern ice margin at $\sim 67^\circ$ N stands out, where other simulations show a much further advanced ice sheet towards the coastline. Also at other locations (north and northeast), the current position of the GrIS margin is best simulated using the "Utr-8" SMB forcing.

## 5   Discussion

The performance of the different albedo schemes is described in terms of the temporal evolution at distinct locations, and in terms of the spatial pattern of $\alpha_{sn}$ in the ablation season. Apart from that, the different albedo schemes are evaluated in terms of their ability to generate a realistic SMB forcing for ice sheet simulations, in contrast to a direct comparison to observed albedo. This is a deliberate choice, which fits in our goal to develop an earth system model that includes interactive ice sheets. In such a model framework, it is an endeavour to simulate a GrIS that remains close to the present-day size under late-Holocene climate

forcing, i.e. the ice sheet should not grow significantly larger, nor should it retreat far from its present-day margin. To this end, we use the albedo parameterization as a calibration tool, since the sensitivity of ice sheet evolution to snow albedo is shown to be large. Our results show that this procedure is able to generate an ice sheet that remains reasonably close to the present-day size. However, we do not claim that our proposed albedo scheme is physically superior to other schemes. Discrepancies remain

with respect to e.g. the spatial distribution of SMB. Due to error compensation, the best parameters as found here make up for model flaws (e.g. in the simplistic snow albedo scheme, but also in the limitations of the ISM) or other unknown flaws, which might lead to errors in future projections. Nevertheless, based on the agreement in ice sheet area and volume, the "Utr-8" albedo parameterization seems the best parameter setting within the set of albedo schemes to be used in EC-Earth simulations with the current snow scheme and an interactive ice sheet component. Adding a more sophisticated snow scheme will likely

change optimal choices for albedo parameters.

Our modifications of the EC-Earth albedo scheme do not yet lead to a SMB forcing field that is of equal quality as produced by climate models using more sophisticated snow schemes (e.g. Vizcaíno et al., 2013; Fettweis et al., 2013; Van Angelen et al., 2014). An important omission of the snow albedo scheme in EC-Earth over ice sheets as described here is that it does not distinguish between melting ice and melting snow. This has in particular implications for the ablation zone, where the seasonal

removal and re-appearance of a snow/firn layer strongly influences the seasonal cycle of the albedo, determining the start of the ablation season. The pattern of underestimated SMB in the lower ablation zone, and overestimated melt in the higher parts (Figure 5h and 6h) is likely partly an effect of this lack of discrimination between wet snow and ice albedo, and no account of the effect of impurity content. Moreover, this effectively means that the effects of liquid water in snow and the (lower) albedo of bare ice with or without impurities are lumped into one exponential function for melting conditions. This may explain our

result that the rapid exponential decay ($\tau_{melt}$=1 day) and the relatively low minimum albedo ($\alpha_{min}$=0.50) provides best results. More adjustments to the snow scheme are necessary to enable a distinction between the bare ice area and the firn-covered ice sheet. This is not feasible for this study, but it is a key recommendation for improvement in the future development of the land surface model.

The snow albedo scheme in EC-Earth is diagnostic, as albedo depends on the time past since last snowfall. This type of

albedo parameterization is of similar complexity as snow albedo schemes in other GCMs (e.g. CNRM, Voldoire et al. (2013); MIROC, Watanabe et al. (2010)). For a further improvement of the description of snow albedo, the snow model needs to include snow grain size evolution, to enable calculation of snow albedo as a function of grain size, such as for example used in CESM (Lipscomb et al., 2013).

Another omission in our SMB calculation is the effect of refreezing of percolating meltwater. Refreezing has a buffering

effect on mass loss by runoff, especially at the equilibrium line and higher. Accordingly, neglecting the effect of refreezing provides an explanation for underestimation of SMB around the equilibrium line in Figure 5h. An extension of the number of layers in the snow model can improve the simulation of refreezing of percolating meltwater. This will considerably decrease the total runoff, and hence improve simulated ice-sheet integrated SMB value compared to contemporary assessments from RCMs. Moreover, accurately describing the effect of refreezing will improve the spatial distribution of SMB, which is of large importance for interactive climate – ice sheet model simulations.

The problem of a difference in horizontal resolution between climate model and ISM is circumvented here by applying a correction to the SMB fields based on the difference in elevation (Helsen et al., 2012). This method is based on the spatially-varying relations between surface elevation and SMB. Similar efforts to parameterize the SMB – elevation feedback exist. Instead of deriving a spatial relation of SMB with elevation, Franco et al. (2012) and Noël et al. (2016) correlate individual SMB components to elevation. Edwards et al. (2014) uses a suite of climate simulations with different GrIS topographies, to derive SMB–elevation gradients. The latter study does not allow spatially-varying relations, apart from distinguishing between north and south Greenland, and above and below the equilibrium line. We applied the method by Helsen et al. (2012) because it offers more spatial variability and can be applied outside the current ice sheet mask, a prerequisite for long-term ISM simulations. An alternative approach to correct for the SMB–elevation feedback effects due to differences in spatial resolution is described by Vizcaíno et al. (2013); Lipscomb et al. (2013), who solve SEB and SMB in different subgrid elevation classes in the CESM land model, and subsequently downscale SMB to the high-resolution ISM grid (see section 1). However, their approach only partly allows energy balance terms to vary with elevation, and keeps incoming radiative terms constant for each climate model grid point. The most physically-sound solution for this issue will come with increasing computer power, allowing climate models to fully solve the energy balance at sufficiently high resolution. Moreover, one-to-one ISM–GCM coupling will most adequately account for the height-mass balance feedback.

## 6 Conclusions

We have extended the albedo parameterization over the GrIS in the earth system model EC-Earth, to replace the constant value of 0.80 over perennial snow in EC-Earth. We applied different exponentially-decaying functions to account for the slow and fast response of $\alpha_{sn}$ in dry and wet conditions, respectively. Our results show that small adjustments to the albedo scheme significantly influence the SMB over Greenland. This in turn affects the ice sheet response, implying consequences for coupled ice sheet – climate simulations in an earth system model framework. The height - mass balance effect that we parameterized here using a relation of SMB with surface elevation will be more accurately solved when ice sheet elevation and extent are given back to the climate model. Based on the ice sheet simulations, the "Utr-8" albedo parameterization seems the most suitable albedo scheme to be used in EC-Earth simulations with an interactive ice sheet component. However, based on the different results obtained with a climate forcing from a RCM, our results emphasize the importance of capturing the spatial distribution of the SMB, rather than the ice-sheet integrated number. We note that the physics of the albedo scheme can still be greatly improved with the inclusion of a multi-layer snow model component in the land surface component of EC-Earth, to better account for refreezing of percolating meltwater in snow, and to distinguish between bare ice and snow. Hence, further improvements of the snow scheme are crucial for the development of earth system models including an interactive ice sheet component.

*Acknowledgements.* MH was supported by The Netherlands Polar Programme (NPP) of the Earth and Life Sciences division of The Netherlands Organisation for Scientific Research (NWO/ALW). TR was financially supported by the Netherlands Earth System Science Centre

(NESSC, NWO/ALW). MSM and SY were partly supported by Nordic Centers of Excellence eSTICC (sScience Tool for Investigating Climate Change in northern high latitudes) and SVALI (Stability and Variations of Arctic Land Ice), funded by Nordforsk ( grant 57001 and grant 24420).

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

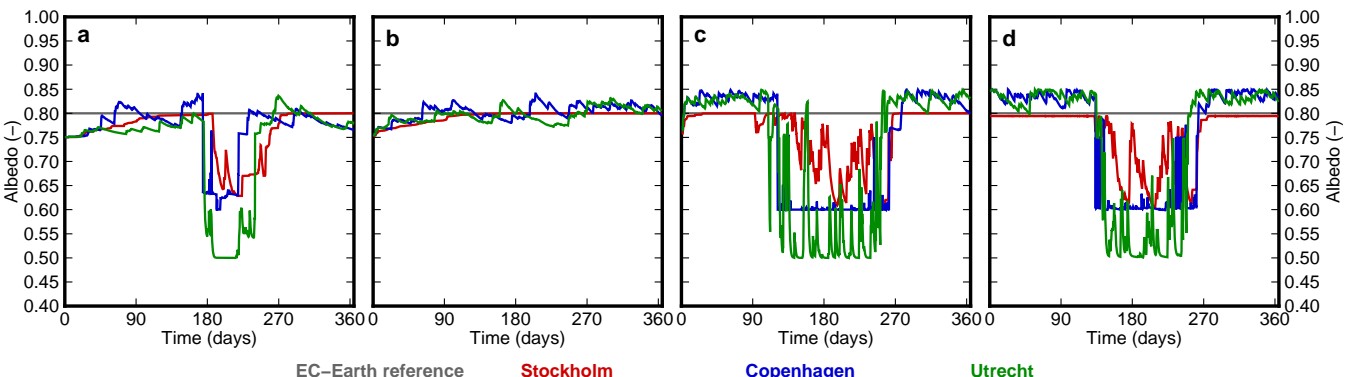

**Figure 1.** Evolution of albedo during one year at four locations over the GrIS: northwestern ablation area (a), Summit (b), southwestern ablation area (c) and southeastern high accumulation area (d). Locations are shown as circles in Figure 2. Albedo schemes "Sto" (red), "Cph" (blue) and "Utr-8" (green). Albedo in the standard version of EC-Earth3 is constant (gray line).

**Table 1.** Albedo scheme parameter values.

| Parameter | Sto | Cph | Utr-5 | Utr-6 | Utr-7 | Utr-8 | Utr-9 |
|---|---|---|---|---|---|---|---|
| $\alpha_{min}$ | 0.60 | 0.60 | 0.50 | 0.50 | 0.50 | 0.50 | 0.45 |
| $\alpha_{refr}$ | - | 0.65 | 0.60 | 0.60 | 0.60 | 0.60 | 0.60 |
| $\alpha_{firn}$ | - | 0.75 | 0.75 | 0.75 | 0.75 | 0.75 | 0.75 |
| $\alpha_{max}$ | 0.80 | 0.85 | 0.80 | 0.85 | 0.85 | 0.85 | 0.80 |
| $\tau_{melt}$ | 4 days | 0 days | 4 days | 4 days | 2 days | 1 day | 0 days |
| $\tau_{firn}$ | - | 30 days | 30 days | 30 days | 30 days | 30 days | 30 days |

**Table 2.** GrIS area-integrated values of SMB components (SF=snowfall, E=evaporation, M=melt, R=runoff, SMB=surface mass balance, all numbers are given in Gt yr$^{-1}$). The uncertainty estimate is the standard deviation from the interannual variability.

| | ref | Sto | Cph | Utr-5 | Utr-6 | Utr-7 | Utr-8 | Utr-9 | RACMO2[*] |
|---|---|---|---|---|---|---|---|---|---|
| SF | 768 ±72 | 736 ±85 | 754 ±69 | 761 ±67 | 768 ±56 | 733 ±63 | 731 ±94 | 762 ±75 | 673 ±76 |
| E | 18 ±4 | 20 ±3 | 21 ±3 | 26 ±2 | 18 ±3 | 23 ±3 | 24 ±4 | 37 ±4 | 41 ±3 |
| M | 270 ±28 | 371 ±56 | 418 ±45 | 460 ±63 | 389 ±59 | 443 ±87 | 480 ±92 | 693 ±165 | 436 ±68 |
| R | - | - | - | - | - | - | - | - | 258 ±49 |
| SMB | 479 ±85 | 345 ±121 | 315 ±88 | 274 ±102 | 361 ±85 | 267 ±123 | 227 ±152 | 32 ±152 | 407 ±111 |
| downscaled-SMB | 327 ±85 | 208 ±121 | 190 ±88 | 159 ±102 | 225 ±85 | 151 ±123 | 115 ±152 | -66 ±152 | 198 ±111 |

[*] RACMO2 data are computed for the period 1960-1989, and are projected to the ice mask on 20 km resolution to ensure a fair comparison, but this causes small differences with the original data reported in Van Angelen et al. (2014)

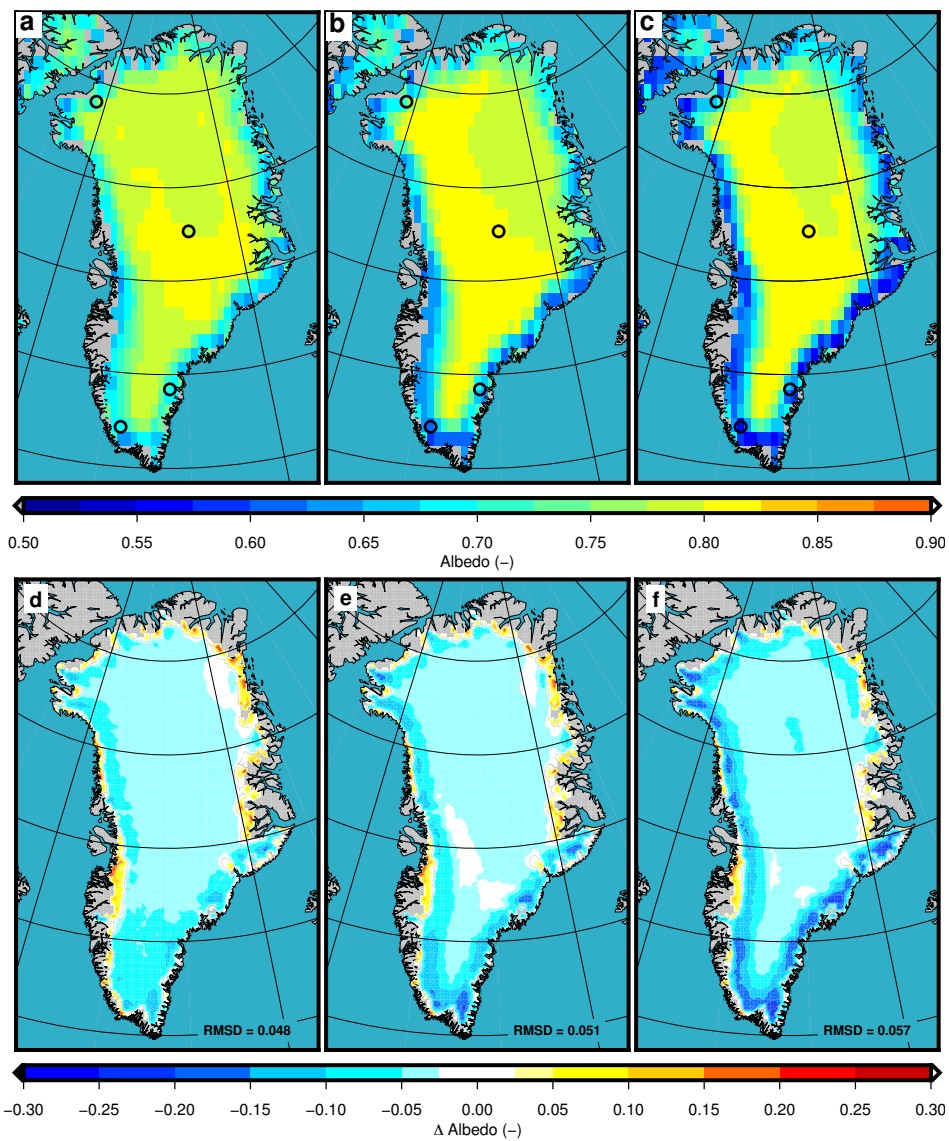

**Figure 2.** Average JJA albedo (1990-2012) from EC-Earth for the "Sto" scheme (a), "Cph" scheme (b) and the "Utr-8" scheme (c). Difference with JJA albedo from RACMO2 (1960-1989, Van Angelen et al., 2014) is shown in (d-f) The circles indicate the locations of the four time series shown in Figure 1. Albedo is only shown for the ice sheet surface type, i.e. grey land area is outside the ice sheet domain in EC-Earth.

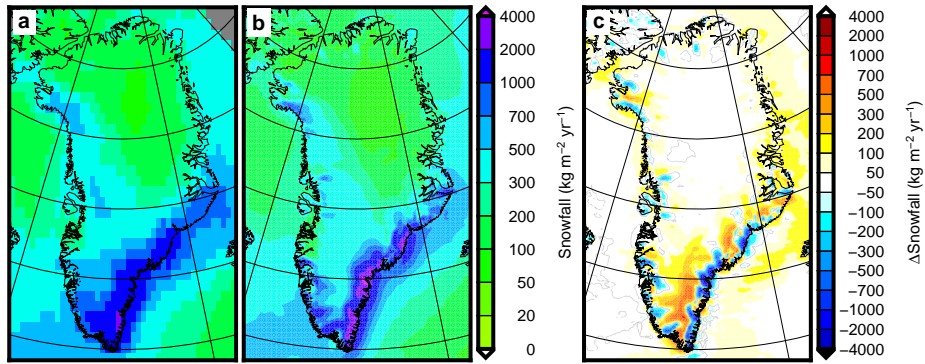

**Figure 3.** Average snowfall (1990-2012) from the reference run (a), from RACMO2 (Van Angelen et al., 2014) (b), and difference (EC-Earth - RACMO) (c).

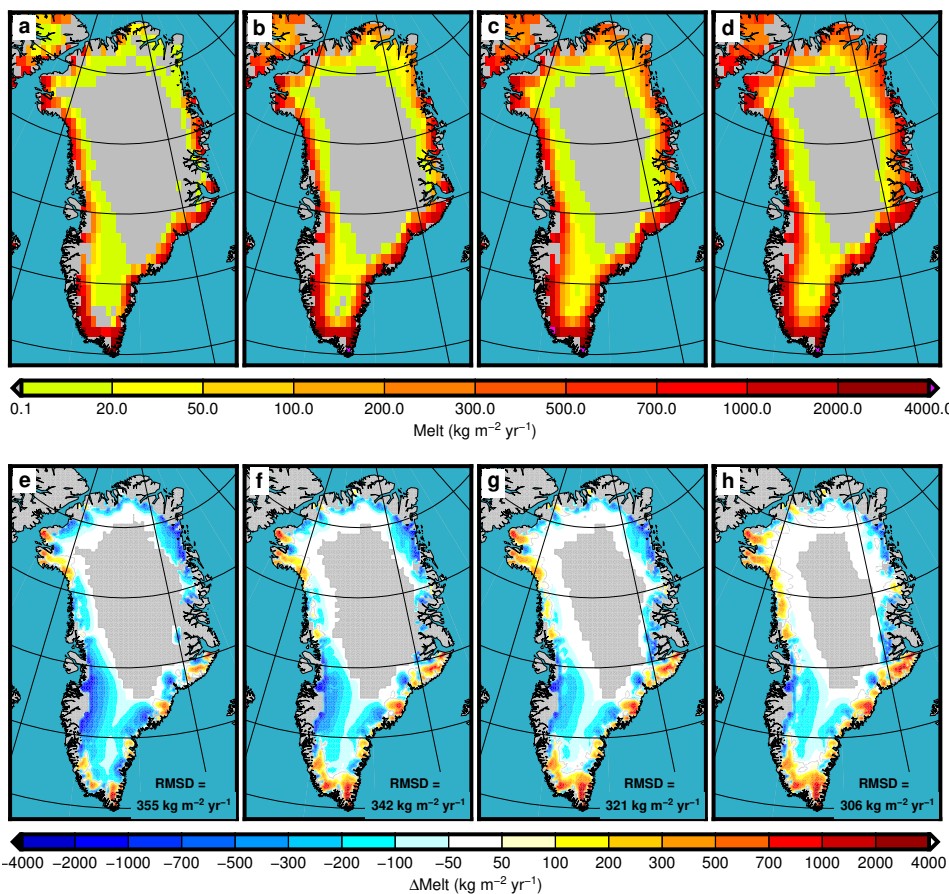

**Figure 4.** Average melt (1990-2012) from the reference run (a) and modified albedo schemes "Sto" (b), "Cph" (c) and "Utr-8" (d). Difference with melt from RACMO2 (Van Angelen et al., 2014) is shown in (e-h).

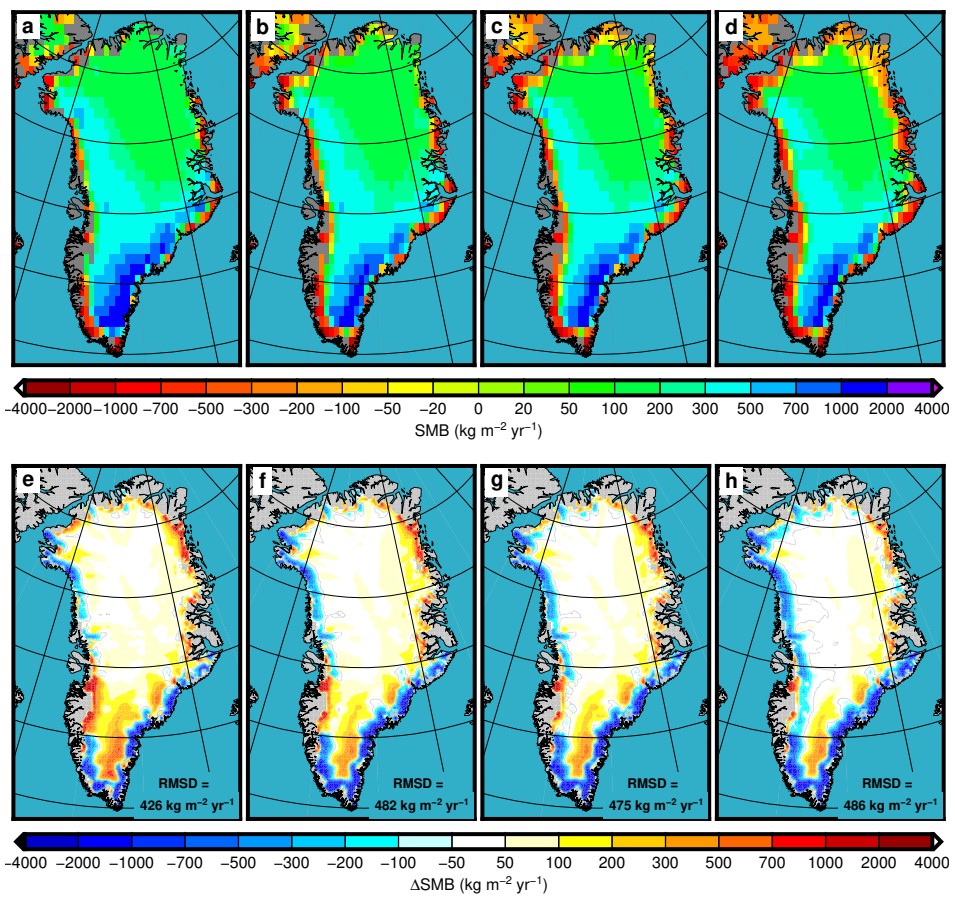

**Figure 5.** Average SMB (1990-2012) from the reference run (a) and from runs using modified albedo schemes "Sto" (b), "Cph" (c) and "Utr-8" (d). Difference with SMB from RACMO2 (Van Angelen et al., 2014) is shown in (e-h).

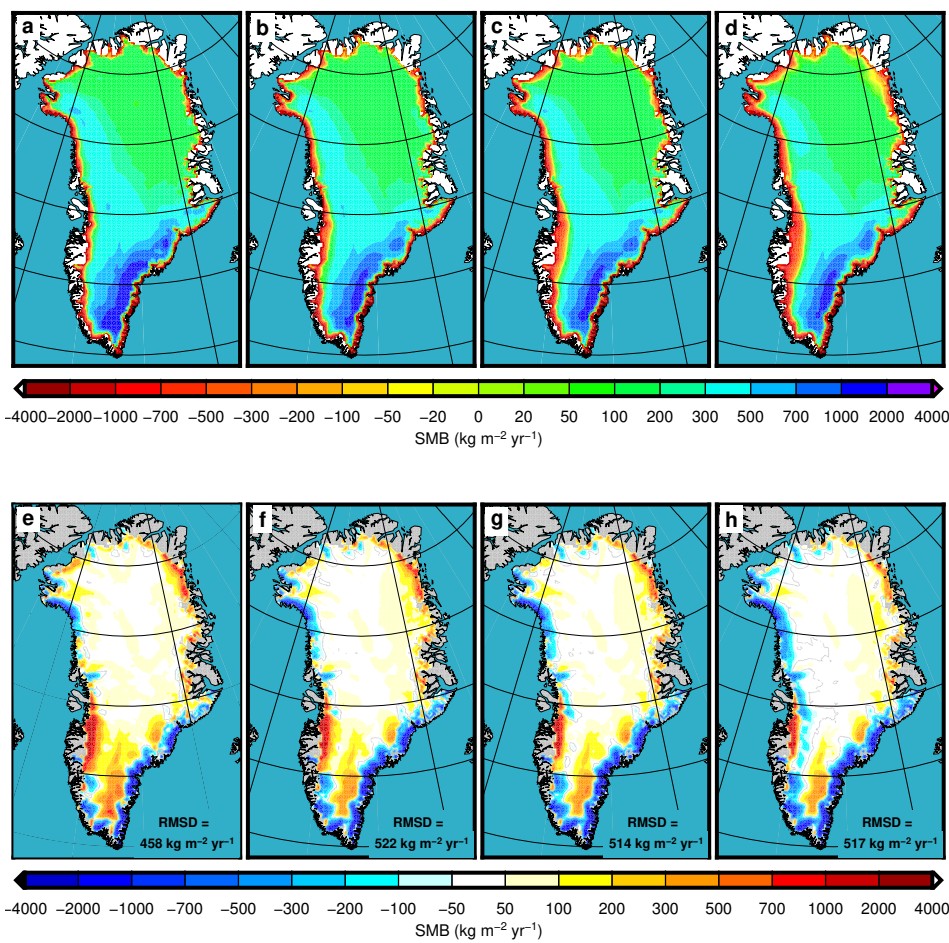

**Figure 6.** Downscaled SMB from the reference run (a) and from runs using albedo schemes "Sto" (b), "Cph" (c) and "Utr-8" (d), and the difference with SMB$_{RACMO2}$ (e-h).

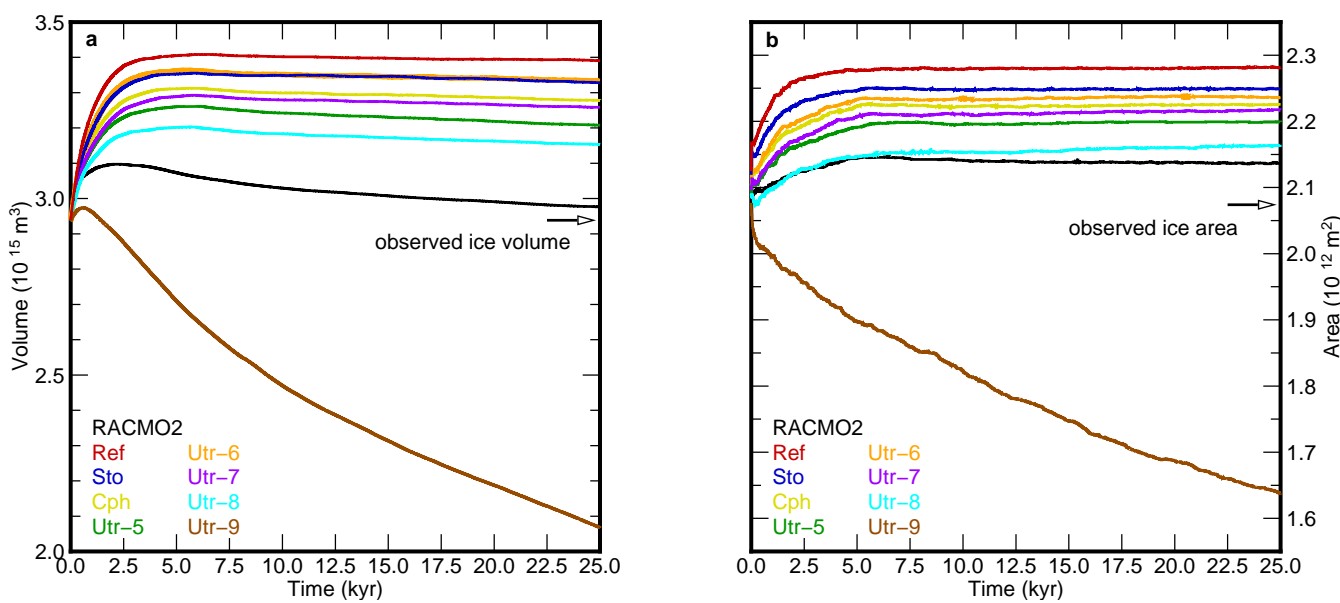

**Figure 7.** Ice volume (a) and area (b) resulting from simulations forced with different climatologies.

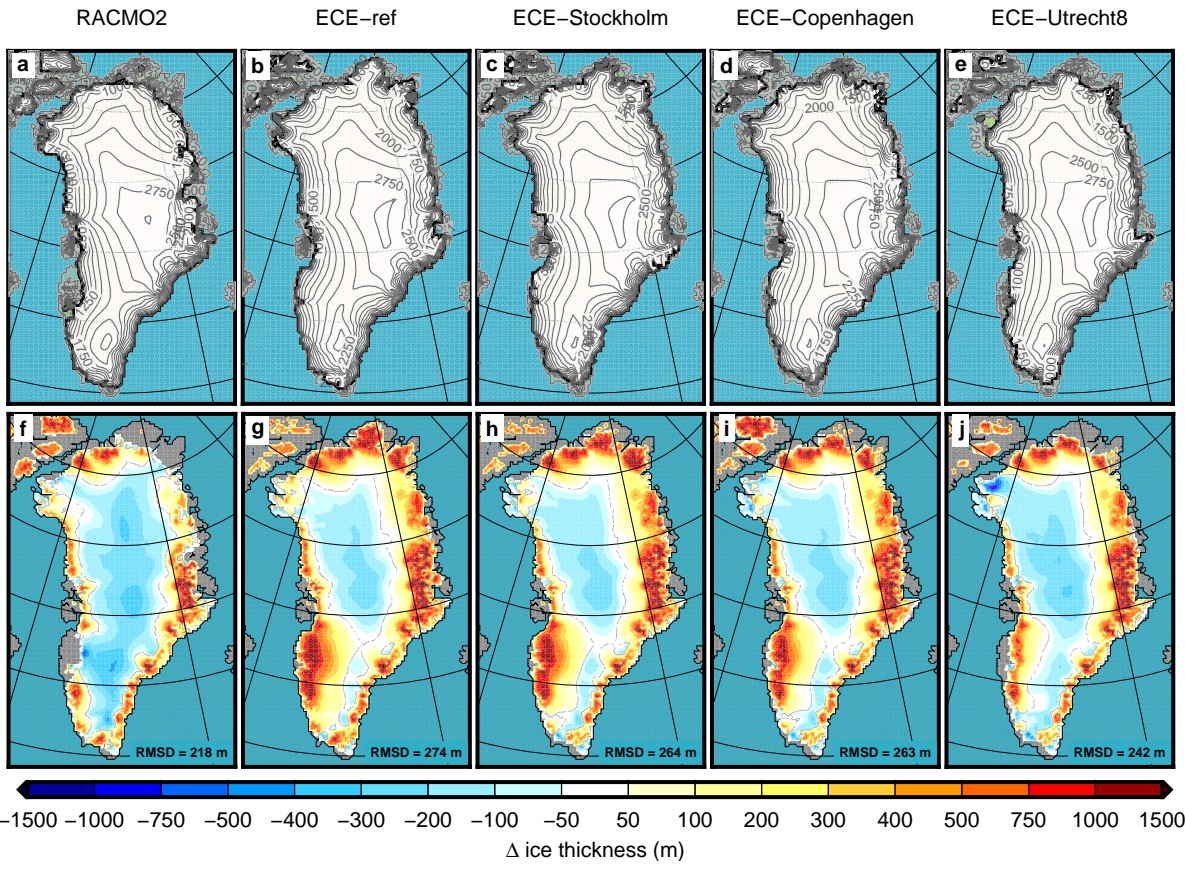

**Figure 8.** Ice sheet surface elevation (panel a-e) and difference of ice thickness with present-day (f-j) resulting from 25,000 yr simulations forced with different climatologies. Root-mean-square deviation (RMSD) is given as a measure for the mismatch between simulated and observed ice thickness.