# Peer review of "On the importance of the albedo parameterization for the mass balance of the Greenland ice sheet in EC-Earth"

_The Cryosphere, 2016_

## Referee Comment (RC1) · X. Fettweis (Referee) · 19 Jan 2017

This paper presents over the Greenland ice sheet sensitivity experiments performed with EC-Earth discussing parameters of simple albedo parametrizations. This paper is rather a "model development" paper (fitting well in GMD) and does not really bring new stuff. However, showing EC-Earth performance over GrIS could deserve to be published in TC. However, before final acceptance, some revisions are needed:

- No validation of albedo with observations is presented. A comparison with the mean RACMO albedo (which is enough robust to be considered as observation) is here at least needed as the melt biases (Fig4) are discussed in fct of albedo biases.

[Figure]

- pg8, lines 9-18: We observe exactly the same differences/biases in the precipitation patterns when MAR is run at lower resolutions as shown in Franco et al. (TC, 2012) who should be cited here.

- No evaluation of the SEB is shown. Again, a comparison of the incoming shortwave and longwave fluxes with RACMO outputs should be added. I know that RACMO (and MAR) have significant biases in simulated SWD/LWD but it is better than nothing. Due to error compensations (as it is the case with MAR), overestimation of SWD can be compensated by too high albedo for example. A comparison of EC-Earth simulated SWD/LWD will able to better interpret comparisons shown in Fig4.

- pg 10, lines 1-2: the apparent decrease of the RACMO SMB is strange and is an artifact of the interpolation. The comparison and statistics listed in Table 1 should be made on the common ice sheet mask and not on the ISM ice sheet mask.

- pg 11, lines 28-31: I fully agree with the authors that a large part of their biases are due to the no-distinction between melting bare ice and melting snow. The melting snow albedo (alpha min) used here is artificially too low in the aim of approximating the bare ice albedo over the ablation zone. The best should be to have two albedo. As the snow model computes SMB, it looks like easy for me to implement a simple correction of the albedo parmetrisation following SMB values. If SMB <0 (or if ablation > winter accumulated snowpack), then alpha min = 0.40 else alphamin=0.6 for example. The accumulated snow height from previous 1st Sep could be used to distinguish accumulation to ablation zone. I think that this simple correction of albedo will improve a lot the comparison with RACMO.

- pg 13, line 4: due to some error compensations, best parameters for albedo are not necessary the best to simulated the present ice sheet topography. This should be clearly mentioned in the text.

- It should be interesting to test the albedo parametrization of SEMIC which seems to be easily to implement. There is here a clear distinction between snow and ice albedo.

Krapp, M., Robinson, A., and Ganopolski, A.: SEMIC: an efficient surface energy and mass balance model applied to the Greenland ice sheet, The Cryosphere Discuss., doi:10.5194/tc-2016-252, in review, 2016.

---

## Referee Comment (RC2) · Anonymous Referee #2 · 21 Jan 2017

**Influence of albedo parameterization on surface mass balance in the perspective of Greenland ice sheet modelling in EC-Earth**

Michiel Helsen, Roderik van de Wal, Thomas Reerink, Richard Bintanja, Marianne Sloth Madsen, Shuting Yang, Qiang Li, and Qiong Zhang

**Summary**:
The authors implement and test the impact of a new, relatively simple ice sheet albedo scheme for the EC-Earth earth system model with respect to Greenland ice sheet surface mass balance. The impact of varying parameters in the scheme on SMB is assessed. SMB is then used to force the IMAU-ICE ice sheet model to understand the impact of the range of SMB simulations on overall ice sheet mass balance.

**General Comments:**
The study is an interesting exploration of the impact of the albedo parameterization on both the surface mass balance in a GCM and the subsequent state of the ice sheet. It shows that small changes in the proposed parameters for a simple albedo parameterization can have a large effect on the state of the ice sheet, as well as the importance of including a dynamic albedo scheme in future projections of ice sheet change in GCMs. Some issues need to be addressed before the paper can be published. In particular:

(1) I agree with Dr. Fettweis that there should be some comparison between simulated and observed albedo values. Despite the authors' statement that the focus of the analysis is not necessarily to achieve an accurate simulation of albedo, it is still useful to know how well the different simulations agree with observed albedo, which would reveal strengths and limitations of the current scheme. This would not necessarily need to be a very detailed evaluation, given the relative simplicity of the scheme employed. As Dr. Fettweis mentioned, RACMO albedo estimates might be sufficient. In particular, it would be useful to see a comparison with observed albedo at specific sites (Fig. 1), to better reveal which scheme best captures the typical evolution of albedo in each region. An ice-sheet wide comparison, such as the comparison shown in Fig. 3 for SMB would also be useful.

(2) The authors only mention the influence of bare ice towards the end of the paper. Since it is an important factor it should be discussed more often and earlier in the paper. The model's apparent inability to simulate bare ice exposure is also a limitation that should be mentioned.

(3) It should be emphasized a bit further that the "best" scheme at present may be partially compensating for biases associated other processes that are not simulated, such as refreezing of meltwater and bare ice exposure. Including these processes would necessitate a simultaneous revision and improvement of the albedo scheme. Also the "best" scheme, if not consistent with observed albedo can produce feedbacks that lead to magnified errors in future simulations.

**Specific Comments:**

1. **Title:** The phrase "in the perspective of Greenland ice sheet modelling" doesn't seem grammatically correct.
2. **P. 1, Lines 1-2:** Bare ice exposure should be mentioned as a factor as well.
3. **P. 1, Line 5:** There are not really "eight snow albedo schemes"; there are is really one scheme that is employed, with varying parameters, and some small adjustments. Please revise.
4. **P. 1, Lines 4-8:** More details need to be provided here; the writing style is rather vague. The main purpose of the study is unclear. The source of ice sheet topography should be mentioned, along with the model used for the future projections. Some specific results should be provided.
5. **P. 1, Line 16:** Explain briefly what features of the ISM could lead to different methods for computing SMB inputs.
6. **P. 2, Lines 26-28:** The role of bare ice exposure in variations in albedo should be mentioned here, along with the influence of impurities, especially over bare ice.
7. **P. 3, Line 15:** Mention sublimation here as well.
8. **P. 3, Line 30:** A brief description of the EC Earth model should be given here, including its use of spherical harmonics.
9. **P. 3, Lines 31-32:** Please provide a few more details about the snow scheme. e.g. it appears that only one layer of snow is simulated. Are heat fluxes through the snowpack calculated? What about snow density? Can any liquid water be stored in the snow?
10. **P. 4, Line 1:** Suggest "linearly decreasing" and "exponentially decreasing" in place of "linear" and exponential, for clarity.
11. **P. 4, Line 8:** This sentence is confusing. How can snow accumulate above the maximum thickness?
12. **P. 4, Line 24:** Note that the description here is of the "base" or "control" scheme that is then modified by each group.
13. **P. 4, Line 25:** What is the threshold for resetting the snow albedo? 1 cm?
14. **P. 4, Lines 28-29:** Please specify the units for $\Delta t$ and F. Does 10 represent 10 millimeters?
15. **P. 4, Lines 32-33:** What is meant by the "flux of meltwater"? Is this the flux of meltwater out of the snowpack? Does it include rainfall events? What is the difference between "internal melting" and "flux of meltwater"…does this mean meltwater is stored in the snowpack?
16. **P. 5, Equations (3) and (4):** In this case I believe $\Delta t$ refers to the total elapsed time, rather than the length of the timestep. Please clarify.
17. **P. 6, Lines 5-9:** The 0.45 value is consistent with bare ice… perhaps mention that here.
18. **P. 6, Line 18:** Change "Shown in this figure" to "Shown in Figure 1" for clarity.
19. **P. 6, Line 26:** Change 'The Copenhagen code' to 'The "Cph" scheme'
20. **P. 6, Line 30:** Which figure or set of figures is being referred to here? Perhaps the results for the other simulations could be briefly discussed and included as supplemental figures?
21. **P. 6, Line 31:** It would be helpful to compare with a figure showing this observed pattern.
22. **P. 7, Lines 11-12:** Again, it would be helpful to see the observed albedo.

23. **P. 8, Lines 4-8:** Provide some more details about RACMO, e.g. reanalysis forcing, spatial resolution.  If comparison with RACMO albedo is included, some details of the albedo scheme should be provided.
24. **P. 8, Lines 31-34:** The influence of bare ice and bare ice albedo should be discussed.
25. **P. 9, Line 29:** Clarify "each climatology".
26. **P. 10, Line 1:** Does regridding of RACMO2 qualify as downscaling?  Isn't the model simulation at a higher resolution?
27. **P. 10, Lines 4-5:** Perhaps change "GrIS model" to "GrIS ISM".
28. **P. 10, Lines 16-20:** Some more details are needed here. How is the forcing applied?  Is a monthly climatological SMB forcing applied to the ice sheet model until it reaches steady state?   The "height-mass balance effect" needs to be explained further.  What is the effect of generating runoff beyond 10 m of accumulation on topography changes?
29. **P. 11, Line 31:** The presence of impurities also likely has an impact on the bias here.
30. **P. 12, Lines 1-3:** Although this is the best scheme for agreement with RACMO, it may not be the most accurate scheme, given that it may be compensating for errors in other schemes.  If this is so, then feedbacks may not be accurately captured, leading to errors in future projections.  Please discuss these potential errors.
31. **P. 12, Lines 4-14:** Note that adjusting the scheme for refreezing may require adjustment of the albedo scheme… i.e. they have to be changed together.  Also perhaps simulating exposure of bare ice should also be a subject for future study.
32. **P. 13, Line 5:** Note caveats associated with this being the optimal scheme.

**Technical Corrections:**
1. **P. 1, Line 20:** Change "generally regarded superior" to "generally regarded as superior"
2. **P. 2, Line 12:** Change "ISM component" to "ISM components"
3. **P. 3, Line 1:** Change "To better understand the ice sheet changes and its interaction…" to "To better understand changes in the mass of the ice sheet and in its interaction…"
4. **P. 5, Line 1:** Suggest changing "can be" to "is".
5. **P. 6, Lines 2-3:** The sentence is repetitive.  Perhaps change to read simply 'For dry conditions, the "Utr" scheme adopts the same choices as the "Cph" scheme.'
6. **P. 7, Line 24:** Change "next to that" to "in addition".
7. **P. 8, Line 21:** Change "(and wide)" to "(and extensive)"
8. **P. 9, Line 28:** Change "tje" to "the".

---

## Referee Comment (RC3) · Anonymous Referee #3 · 27 Jan 2017

This study explores the simulation of Greenland surface mass balance (SMB) and ice sheet elevation in the EC-Earth model, associated with different snow albedo parameters. In particular, the parameters that are varied in this study determine maximum and minimum snow albedo, and the rates of albedo decay with time following snowfall in dry and wet snow. The study also determines a set of albedo parameters that lead to an optimal simulation of present-day ice sheet thickness, an exercise that is motivated by the fact that albedo has a strong impact on simulated SMB and therefore ice thickness. I commend the authors for preparing a very well-written manuscript and helpful discussion. I describe a few issues below, however, that I believe warrant further consideration before the paper should be published in TC.

**Major issues:**

- (1) I appreciate that reasonable simulation of ice thickness is critical for an Earth System model, but I do not think SMB and ice thickness performance should be the only criteria used to evaluate snow albedo schemes. The authors identify other important processes that can strongly affect SMB, such as the refreezing of meltwater. This particular process is neglected in the current version of the model, so I am concerned that once this process is accounted for, the albedo scheme determined to be optimal in this study will no longer be optimal. The same argument applies for the improvement of other factors. For example, improvement of model snowfall will also affect simulated SMB and could, theoretically, alter the snow albedo parameters that produce the best ice thickness simulation. These points lead to the following suggestion: Simulated snow albedo from the different schemes should be evaluated against observed albedo. The authors provide a very rough comparison with ranges of albedo measurements on p.7 line 8, but I think a spatial evaluation against observations from a space-borne sensor like MODIS would be much more useful. Although it is interesting and important to know how the different albedo schemes affect simulated SMB and ice thickness, it is also critical, in my opinion, to know how the albedo scheme performs in terms of albedo.
- (2) Related to (1): The determination of optimal albedo parameters seems flawed because, although these parameters produce the lowest RMSD in ice thickness (figure 8), they clearly do not produce the best simulation of SMB (in comparison with RACMO), as shown in Table 2. Thus, I am concerned that these albedo parameters produce the best simulation of ice thickness for the wrong reasons.
- (3) The authors note (p. 6,21) that "Since the simulated climate is freely evolving after initialization, a direct comparison with observed time series is not meaningful." Could land-only simulations be performed, i.e., with fixed atmosphere conditions? Such simulations would enable a comparison of the albedo schemes that is much less affected by weather/climate variability, and would enable a meaningful comparison with observed
time series.

Minor comments:

p.3,1: "its interaction" -> "their interactions"

p.4,33: What is the justification for choosing -2K as the threshold for applying "wet" snow conditions? Are the results at all sensitive to this temperature choice?

Sections 2.2.1 and 2.2.2: The Stockholm and Copenhagen schemes seem unrealistic because they, respectively, neglect dry snow aging and prescribe sharp, instantaneous changes to albedo under wet conditions, e.g., when snow temperature exceeds -2K. I believe these experiments are still useful because they represent bounds, but I suggest acknowledging a bit more clearly that they should not be viewed as realistic albedo schemes and are instead used for boundary-case sensitivity studies. I would also note that abrupt binary changes in physical parameters are often undesirable in climate models because they can lead to instabilities.

p.6 31: "JJA mean pattern" - pattern of what? Please specify.

p.7,25: Why use the skin temperature to determine conditions for snow melting to occur? Is snow temperature not prognosed in the model?

p.8,24: "A side effect of the use of a time-evolving albedo scheme is an increase of interannual variability in melt..." - Yes, but this should be a more realistic feature! (I mention this because "side effect" often connotes an adverse unintended consequence).

p.9,28: "tje"

p.10,16: How are the (relatively short) SMB fields applied in the 25-ky runs? Are the multi-year SMB timeseries (only 25 years in length, I think) simply repeated? Or are they averaged and repeated annually?

p.10,17: "A uniform lapse rate of -7.4 K km-1 is applied." - Why is this lapse rate used? It seems steeper than values commonly used in other downscaling studies.

**TCD**
p.11,1: "downscaling the locally strongly negative SMB values are" - Please fix for clarity.

p.13,2-4: "The height - mass balance..." - This sentence (particularly the reference to "given back to the climate model") is somewhat unclear to me.

Figure 6: I suggest showing RMSD statistics for the SMB in this figure, as shown in figure 8 for ice thickness.

---

## Author Comment (AC1) · 23 May 2017

The comment was uploaded in the form of a supplement:
http://www.the-cryosphere-discuss.net/tc-2016-281/tc-2016-281-AC1-supplement.pdf

---

## Author Comment (AC2) · 23 May 2017

**Response to reviewer #1**

*No validation of albedo with observations is presented. A comparison with the mean RACMO albedo (which is enough robust to be considered as observation) is here at least needed as the melt biases (Fig4) are discussed in fct of albedo biases.*

To evaluate simulated albedo, we followed the suggestion of the reviewer and chose to compare simulated EC-Earth June-July-August (JJA) albedo with JJA RACMO2 albedo. Figure 2 has been revised, and now includes 3 more panels in which the bias with RACMO2 albedo is plotted:

[Figure]

Revised Figure 2: Average JJA albedo (1990-2012) from EC-Earth for the "Sto" scheme (a), "Cph" scheme (b) and the "Utr-8" scheme (c). Difference with JJA albedo from RACMO2 (Van Angelen et al., 2014) is shown in (d-f) The circles indicate the locations of the four time series shown in Figure 1. Albedo is only shown for the ice sheet surface type, i.e. grey land area is outside the ice sheet domain in EC-Earth.

The following paragraph discussing this evaluation was added (P7,L30-P8,L9):

For a spatial comparison, we compare our simulated JJA albedo fields with albedo from the high-resolution regional atmospheric climate model RACMO2 (1960-1989) (Van Angelen et al., 2014). Forced with ERA-40 (Uppala et al., 2005) and ERA-interim (Simmons et al., 2007) at its lateral boundaries, RACMO2 is run on a 11 km horizontal resolution, and it has 40 sigma levels in the vertical. To facilitate a comparison, EC-Earth fields and RACMO2 fields are mapped onto a 20 km rectangular grid (also used for the ISM simulations, see below). The difference with JJA albedo from RACMO2 is plotted in Figure 2d-f. This comparison reveals that the different schemes in EC-Earth slightly underestimate albedo in the upper accumulation area, but biases are small (<0.05). The bias becomes larger in a narrow zone in the lower accumulation area, where EC-Earth JJA albedo is lower than RACMO2 albedo. This is a firn-covered area of the ice sheet, where our snow albedo scheme falls short, as it does not distinguish between snow and ice. Hence melt conditions lead to (too) fast declining albedo values, leading to slightly deteriorated RSMD values. Instead, snow albedo in RACMO2, based on effective snow grain size in a more sophisticated snow scheme (Kuipers Munneke et al., 2011), remains higher for melt conditions in this area. However, in the lower ablation zone the bias becomes positive, and these biases are smallest for the Utr-8 scheme. Note that RACMO2 uses observed (MODIS) albedo in case the snow layer is absent, so for this region we effectively compare with satellite-observed albedo. This positive bias in the lower ablation zone indicates that summertime albedo can occasionally drop to lower values than 0.5 (minimum snow albedo used in this study).

*- pg8, lines 9-18: We observe exactly the same differences/biases in the precipitation patterns when MAR is run at lower resolutions as shown in Franco et al. (TC, 2012) who should be cited here.*

OK, thanks for the suggestion, we included this citation.

**- No evaluation of the SEB is shown. Again, a comparison of the incoming shortwave and longwave fluxes with RACMO outputs should be added. I know that RACMO (and MAR) have significant biases in simulated SWD/LWD but it is better than nothing. Due to error compensations (as it is the case with MAR), overestimation of SWD can be compensated by too high albedo for example. A comparison of EC-Earth simulated SWD/LWD will able to better interpret comparisons shown in Fig4.**

We think that an evaluation of the SEB is beyond the scope of this manuscript, since it requires many additional figures of fluxes of shortwave and longwave radiation, sensible and latent heat. This would be a too large change of the subject of the manuscript.

However, as suggested we did a comparison of simulated incoming shortwave and longwave radiation between EC-Earth and RACMO2, which gave broadly comparable fields. To illustrate this, we do include these plots in this reply, but chose to leave them out of the manuscript and address the outcome of the comparison only qualitatively in the text.

[Figure]

*Figure Rebuttal 1: Incoming shortwave radiation (upper panels) from EC-Earth, RACMO2, and the difference (EC-Earth – RACMO2); Incoming longwave radiation (lower panels) from EC-Earth, RACMO and the difference (EC-Earth – RACMO2).*

Due to the higher resolution, SWd in RACMO2 contains more spatial variability, especially near the ice edge and over rugged coastlines, whereas EC-Earth shows a smoother field. Differences were mostly within 10 W m$^{-2}$ over the ice mask.
Over most of the ice sheet surface, downwelling longwave radiation (LWd) is slightly higher in EC-Earth than in RACMO2. This difference becomes larger over the ice marginal area, suggesting warmer air over the ice sheet in EC-Earth.
The patterns of the differences in SWd and LWd do not suggest that they play a key role in explaining the pattern of differences in melt (figure 4 in the manuscript). This point is relevant for the manuscript, so we included a few lines on this subject in the discussion (P9,L16-21):

The differences in melt rate between EC-Earth and RACMO2 may also be influenced by biases in incoming radiation. However, a comparison of incoming shortwave and longwave radiation between EC-Earth and RACMO2 reveals that the differences in incoming shortwave radiation are mostly small over the ice sheet. Downwelling longwave radiation is slightly higher in EC-Earth than in RACMO2, suggesting warmer air over the ice sheet in EC-Earth. This difference is most pronounced over the ice marginal area. From this we conclude that biases

in incoming radiation do not play an important role in explaining the pattern of the difference in melt rate.

**- pg 10, lines 1-2: the apparent decrease of the RACMO SMB is strange and is an artifact of the interpolation. The comparison and statistics listed in Table 1 should be made on the common ice sheet mask and not on the ISM ice sheet mask.**

We agree with the reviewer that comparisons should be made on the common ice mask. In fact, this is exactly what we did. Due to the different resolutions used in the climate models, we have to map all results to the ISM ice mask, for a fair comparison. This is why also the RACMO2 fields are interpolated to the SMB ice mask (of 20x20km resolution).

**- pg 11, lines 28-31: I fully agree with the authors that a large part of their biases are due to the no-distinction between melting bare ice and melting snow. The melting snow albedo (alpha min) used here is artificially too low in the aim of approximating the bare ice albedo over the ablation zone. The best should be to have two albedo. As the snow model computes SMB, it looks like easy for me to implement a simple correction of the albedo parmetrisation following SMB values. If SMB <0 (or if ablation > winter accumulated snowpack), then alpha min = 0.40 else alphamin=0.6 for example. The accumulated snow height from previous 1st Sep could be used to distinguish accumulation to ablation zone. I think that this simple correction of albedo will improve a lot the comparison with RACMO.**

We agree with the reviewer that this will probably result in an improvement of performance of the ice sheet albedo scheme. Nevertheless, it is not feasible at this point to make these adjustments in the albedo scheme in EC-Earth. It would require recomputing and recalibrating all of our results. This also bears the risk of overtuning to the observations by basically adopting the hybrid approach where the bare ice albedo is not modeled anymore but replaced by satellite data as done in RACMO2.
The albedo parameterization will obviously have a top priority in future model development for EC-Earth and we foresee that a more sophisticated albedo scheme may by then be a more rigorous solution.

**-pg 13, line 4: due to some error compensations, best parameters for albedo are not necessary the best to simulated the present ice sheet topography. This should be clearly mentioned in the text.**

We rephrased a paragraph in the Discussion section about this subject (P12,L22-25):

However, we do not claim that our proposed albedo scheme is physically superior to other schemes. Due to error compensation, the best parameters as found here make up for other unknown model flaws, which  might lead to errors in future projections. Nevertheless, based on the ice sheet simulations, the "Utr-8" albedo parameterization seems the best parameter setting within the set of albedo schemes to be used in EC-Earth simulations with an interactive ice sheet

component.

**- It should be interesting to test the albedo parametrization of SEMIC which seems to be easily to implement. There is here a clear distinction between snow and ice albedo.**

**Krapp, M., Robinson, A., and Ganopolski, A.: SEMIC: an efficient surface energy and mass balance model applied to the Greenland ice sheet, The Cryosphere Discuss., doi:10.5194/tc-2016-252, in review, 2016.**

We thank the reviewer for this citation, and we will consider this for a new revision whenever a more sophisticated albedo scheme will be implemented.

**Response to reviewer #2**

**(1) I agree with Dr. Fettweis that there should be some comparison between simulated and observed albedo values. Despite the authors' statement that the focus of the analysis is not necessarily to achieve an accurate simulation of albedo, it is still useful to know how well the different simulations agree with observed albedo, which would reveal strengths and limitations of the current scheme. This would not necessarily need to be a very detailed evaluation, given the relative simplicity of the scheme employed. As Dr. Fettweis mentioned, RACMO albedo estimates might be sufficient. In particular, it would be useful to see a comparison with observed albedo at specific sites (Fig. 1), to better reveal which scheme best captures the typical evolution of albedo in each region. An ice-sheet wide comparison, such as the comparison shown in Fig. 3 for SMB would also be useful.**

We chose to include a comparison with albedo from RACMO2. As such, we included an ice sheet wide comparison, as the reviewer suggested. We did not include a comparison of albedo time series (such as shown in Fig.1) with observed albedo, since our simulated climate is evolving freely.
For more details about the comparison of albedo with RACMO2 albedo, see response to reviewer #1.

**(2) The authors only mention the influence of bare ice towards the end of the paper. Since it is an important factor it should be discussed more often and earlier in the paper. The model's apparent inability to simulate bare ice exposure is also a limitation that should be mentioned.**

Yes, we agree. We have included the following line in Section 2.2 "Adjustments for ice sheets" (P5,L20):

Note that we do not make an explicit distinction between albedo of snow and ice. When all snow is melted on a grid point within the ice sheet mask, we use a constant minimum albedo ($\alpha_{min}$).

**(3) It should be emphasized a bit further that the "best" scheme at present may be partially compensating for biases associated other processes that are not simulated, such as refreezing of meltwater and bare ice exposure. Including these processes would necessitate a simultaneous revision and improvement of the albedo scheme. Also the "best" scheme, if not consistent with observed albedo can produce feedbacks that lead to magnified errors in future simulations.**

Yes, we agree. As such, we added the following in the Discussion section (P12, L22-25):

We do not claim that our proposed albedo scheme is physically superior to other schemes. Due to error compensation, the best parameters as found here make up for other unknown model flaws, which might lead to errors in future projections.

Nevertheless, based on the ice sheet simulations, the "Utr-8" albedo parameterization seems the best parameter setting within the set of albedo schemes to be used in EC-Earth simulations with an interactive ice sheet component.

**Specific Comments:**

**1. Title: The phrase "in the perspective of Greenland ice sheet modelling" doesn't seem grammatically correct.**

We modified the title to: On the importance of the albedo parameterization for the surface mass balance of the Greenland ice sheet in EC-Earth

**2. P. 1, Lines 1-2: Bare ice exposure should be mentioned as a factor as well.**

Yes, done.

**3. P. 1, Line 5: There are not really "eight snow albedo schemes"; there are is really one scheme that is employed, with varying parameters, and some small adjustments. Please revise.**

Modified

**4. P. 1, Lines 4-8: More details need to be provided here; the writing style is rather vague. The main purpose of the study is unclear. The source of ice sheet topography should be mentioned, along with the model used for the future projections. Some specific results should be provided.**

We state the main purpose of the study more clear (P1, L5-6):

The purpose of this study is to improve the SMB forcing of the GrIS, by evaluating different parameter settings within a snow albedo scheme.

We don't think that detailed information such as the source of the ice sheet topography is appropriate information for the abstract. Furthermore, we do not make any future projections, so this point is probably a misunderstanding of the reviewer.

**5. P. 1, Line 16: Explain briefly what features of the ISM could lead to different methods for computing SMB inputs.**

We added (P1, L17-18):
This choice is usually determined by the nature of the ISM simulation: future projections rely on climate model output, where observations of snow accumulation can be used for present-day analysis.

**6. P. 2, Lines 26-28: The role of bare ice exposure in variations in albedo should be mentioned here, along with the influence of impurities, especially over bare ice.**

Done (P2, L29-31):

In the absence of a snow layer, albedo of bare ice is much lower, and depends on impurity content (Greuell and Genthon, 2004; Bøggild et al., 2010; Wientjes and Oerlemans, 2010; Wientjes et al., 2011).

**7. P. 3, Line 15: Mention sublimation here as well.**

Done

**8. P. 3, Line 30: A brief description of the EC Earth model should be given here, including its use of spherical harmonics.**

We moved this description to this point (P4, L4-7):

EC-Earth (version 2.3) participated in CMIP5 (Taylor et al., 2012), and its performance was good, in terms of the mean state, 5 spatial patterns, seasonal cycle and variability of present-day climate (Hazeleger et al., 2010, 2012). We use the updated EC- Earth version 3.1, i.e., ECMWF's Integrated Forecasting System (IFS) on a spectral resolution truncated at wave number 255, with 91 vertical atmospheric levels.

**9. P. 3, Lines 31-32: Please provide a few more details about the snow scheme. e.g. it appears that only one layer of snow is simulated. Are heat fluxes through the snowpack calculated? What about snow density? Can any liquid water be stored in the snow?**

Indeed, only one layer of snow is simulated. We included the following lines to provide more details about the snow scheme (P4, L8-10):

… it includes an explicit snow scheme, which consists of one layer of snow, with a time-evolving density (Dutra et al., 2010). Heat fluxes are calculated through the snow layer, and snow liquid water capacity is approximated as a function of density and snow mass.

**10.     P. 4, Line 1: Suggest "linearly decreasing" and "exponentially decreasing" in place of "linear" and exponential, for clarity.**

Done

**11.     P. 4, Line 8: This sentence is confusing. How can snow accumulate above the maximum thickness?**

The maximum thickness is a switch parameter set in the climate model to avoid undesirable results and is not a physical parameter.

To avoid confusion, we rewrote this sentence as (P4, L18-19):

When snowfall occurs on top of a snow layer with maximum thickness, the excess snow is returned to the hydrological cycle as runoff, i.e. by adding it to the meltwater flux.

**12.     P. 4, Line 24: Note that the description here is of the "base" or "control" scheme that is then modified by each group.**

We think this is adequately described in the text as it is.

**13.     P. 4, Line 25: What is the threshold for resetting the snow albedo? 1 cm?**

The value for F is 1 kg m$^{-2}$ h$^{-1}$, which means that 10 kg m$^{-2}$ of fresh snow is needed to reset the snow albedo to its maximum value.

**14.     P. 4, Lines 28-29: Please specify the units for t and F. Does 10 represent 10 millimeters?**

Yes. The unit is for F is kg m$^{-2}$ h$^{-1}$, which indeed means that 10 kg m$^{-2}$ will reset the snow albedo.

**15.     P. 4, Lines 32-33: What is meant by the "flux of meltwater"? Is this the flux of meltwater out of the snowpack? Does it include rainfall events? What is the difference between "internal melting" and "flux of meltwater"...does this mean meltwater is stored in the snowpack?**

What is meant here is the presence of liquid water, so it includes rainfall events. It is rephrased as follows (P5, L9-11):

So-called "wet" conditions are recognised when either liquid water is present (due to melting or rainfall), or when snow temperature is within 2K of the melting point.

**16.     P. 5, Equations (3) and (4): In this case I believe t refers to the total elapsed time, rather than the length of the timestep. Please clarify.**

No, $\Delta t$ refers to the time step. This equation calculates the new snow albedo after one time step: the e-folding term becomes larger with a larger time step, and is multiplied with the difference between actual and minimum albedo. This term is then added to the minimum albedo, which gives albedo in the next time step.

**17.	P. 6, Lines 5-9: The 0.45 value is consistent with bare ice... perhaps mention that here.**

Done

**18.	P. 6, Line 18: Change "Shown in this figure" to "Shown in Figure 1" for clarity.**

Done

**19.	P. 6, Line 26: Change 'The Copenhagen code' to 'The "Cph" scheme'**

Done

**20.	P. 6, Line 30: Which figure or set of figures is being referred to here? Perhaps the results for the other simulations could be briefly discussed and included as supplemental figures?**

We refer to all figures in the manuscript. We changed the line such that we now explicitly state this. The results of the other simulations are given in Table 2, but the results are not very different compared to the other simulations. Hence, we don't think the addition of supplemental figures adds much.

**21.	P. 6, Line 31: It would be helpful to compare with a figure showing this observed pattern.**

We revised this statement, as the revised Figure 2 shows that it is not so obvious that the Utr-8 scheme compares best with RACMO2 albedo.

**22.	P. 7, Lines 11-12: Again, it would be helpful to see the observed albedo.**

We refer (in text) to typical values of observed albedo.

**23.	P. 8, Lines 4-8: Provide some more details about RACMO, e.g. reanalysis forcing, spatial resolution. If comparison with RACMO albedo is included, some details of the albedo scheme should be provided.**

We added the requested information, but added this earlier in the text, in section 2.3 where albedo is compared with RACMO2 (P7, L30 – P8, L9):

For a spatial comparison, we compare our simulated JJA albedo fields with albedo from the high-resolution regional atmospheric climate model RACMO2 (1960-1989) (Van Angelen et al., 2014). Forced with ERA-40 (Uppala et al., 2005) and ERA-interim (Simmons et al., 2007) at its lateral boundaries, RACMO2 is run on a 11 km horizontal resolution, and it has 40 sigma levels in the vertical. To facilitate a comparison, EC-Earth fields and RACMO2 fields are mapped onto a

20 km rectangular grid (also used for the ISM simulations, see below). The difference with JJA albedo from RACMO2 is plotted in Figure 2d-f. This comparison reveals that the different schemes in EC-Earth slightly underestimate albedo in the upper accumulation area, but biases are small (<0.05). The bias becomes larger in a narrow zone in the lower accumulation area, where EC-Earth JJA albedo is lower than RACMO2 albedo. This is a firn-covered area of the ice sheet, where our snow albedo scheme falls short, as it does not distinguish between snow and ice. Hence melt conditions lead to (too) fast declining albedo values, lead- ing to slightly deteriorated RSMD values. Instead, snow albedo in RACMO2, based on effective snow grain size in a more sophisticated snow scheme (Kuipers Munneke et al., 2011), remains higher for melt conditions in this area. However, in the lower ablation zone the bias becomes positive, and these biases are smallest for the Utr-8 scheme. Note that RACMO2 uses observed (MODIS) albedo in case the snow layer is absent, so for this region we effectively compare with satellite-observed albedo. This positive bias in the lower ablation zone indicates that summertime albedo can occasionally drop to lower values than 0.5 (minimum snow albedo used in this study).

**24.      P. 8, Lines 31-34: The influence of bare ice and bare ice albedo should be discussed.**

We discuss this influence in the Discussion section.

**25.      P. 9, Line 29: Clarify "each climatology".**

"climatology" is replaced with "SMB field".

**26.      P. 10, Line 1: Does regridding of RACMO2 qualify as downscaling? Isn't the model  simulation at a higher resolution?**

Yes, we agree, so we now use the word "regridding".

**27.      P. 10, Lines 4-5: Perhaps change "GrIS model" to "GrIS ISM".**

Done

**28.      P. 10, Lines 16-20: Some more details are needed here. How is the forcing applied? Is a  monthly climatological SMB forcing applied to the ice sheet model until it reaches steady state? The "height-mass balance effect" needs to be explained further. What is the effect of generating runoff beyond 10 m of accumulation on topography changes?**

No, we do not apply a monthly forcing, but an annual mean SMB value, that follows from our EC-Earth runs. However, for each grid point we adjust the SMB based on the elevation difference between the actual ISM elevation and the

elevation used in the climate model. We multiply this elevation difference with a so-called SMB gradient, which we derive from a correlation between surface elevation and SMB. For this correlation we use all grid points within a distance of 150 km. For an extensive description of this method we refer to Helsen et al. (2012). We added the following text on this subject in the manuscript (P11, L16-20):

The influence of topography changes on SMB (height-mass balance effect) is parameterized by calculating a new SMB forcing field each time step using SMB gradients (Helsen et al., 2012). These gradients are computed for each grid point, from a correlation between surface elevation and SMB values in an area with a radius of 150 km. This allows the ice sheet to advance outside the initial ice sheet mask, but also can lead to substantial retreat when ice sheet thinning occurs (Helsen et al., 2012).

There is no direct effect of runoff on locations with more than 10 m of accumulation. At least, no direct effect in the climate model. It should be noted that these ISM simulations are not carried out within EC-Earth, so we only use SMB fields derived from EC-Earth as a forcing for the ISM. Within the ISM, we do not distinguish between snow or ice; the simulated ice sheet is assumed to be entirely ice. Hence, SMB forcing is applied in terms of ice equivalent, and corrections for the height-mass balance effect is only done within the ISM simulation, and has no implications for EC-Earth.

**29.    P. 11, Line 31: The presence of impurities also likely has an impact on the bias here.**

Yes, we rephrase it as follows (P12, L31 – P13, L1):

The pattern of underestimated SMB in the lower ablation zone, and overestimated melt in the higher parts (Figure 5h and 6h) is likely partly an effect of this lack of discrimination between wet snow and ice albedo, and no account of the effect of impurity content. Moreover, this effectively means that the effects of liquid water in snow and the (lower) albedo of bare ice with or without impurities are lumped into one exponential function for melting conditions.

**30.    P. 12, Lines 1-3: Although this is the best scheme for agreement with RACMO, it may  not be the most accurate scheme, given that it may be compensating for errors in other schemes. If this is so, then feedbacks may not be accurately captured, leading to errors in future projections. Please discuss these potential errors.**

We added the following in the first paragraph of the discussion (P12, L22-24):

However, we do not claim that our proposed albedo scheme is physically superior to other schemes. Due to error compensation, the best parameters as found here make up for other unknown model flaws, which might lead to errors in future projections.

31.      **P. 12, Lines 4-14: Note that adjusting the scheme for refreezing may require adjustment of the albedo scheme... i.e. they have to be changed together. Also perhaps simulating exposure of bare ice should also be a subject for future study.**

Yes, we agree with the reviewer, and take this as a valuable suggestion for future development of the model.

**32.      P. 13, Line 5: Note caveats associated with this being the optimal scheme.**

Yes, we now end the conclusion section with the following sentence (P14, L9-11):

However, we note that the realism of the albedo scheme can still be greatly improved with the inclusion of a multi-layer snow model component in the land surface component of EC-Earth.

**Technical Corrections:**

**1.P. 1, Line 20: Change "generally regarded superior" to "generally regarded as superior"**

Done

**2.P. 2, Line 12: Change "ISM component" to "ISM components"**

Done

**3.P. 3, Line 1: Change "To better understand the ice sheet changes and its interaction..." to  "To better understand changes in the mass of the ice sheet and in its interaction..."**

Done

**4.P. 5, Line 1: Suggest changing "can be" to "is".**

Done

**5.P. 6, Lines 2-3: The sentence is repetitive. Perhaps change to read simply 'For dry  conditions, the "Utr" scheme adopts the same choices as the "Cph" scheme.'**

Done

**6.P. 7, Line 24: Change "next to that" to "in addition".**

Done

**7.P. 8, Line 21: Change "(and wide)" to "(and extensive)"**

Done

**8.P. 9, Line 28: Change "tje" to "the".**

Done

**Response to reviewer #3**

**Major issues:**

**(1) I appreciate that reasonable simulation of ice thickness is critical for an Earth System model, but I do not think SMB and ice thickness performance should be the only criteria used to evaluate snow albedo schemes. The authors identify other important processes that can strongly affect SMB, such as the refreezing of meltwater. This particular process is neglected in the current version of the model, so I am concerned that once this process is accounted for, the albedo scheme determined to be optimal in this study will no longer be optimal. The same argument applies for the improvement of other factors. For example, improvement of model snowfall will also affect simulated SMB and could, theoretically, alter the snow albedo parameters that produce the best ice thickness simulation. These points lead to the following suggestion: Simulated snow albedo from the different schemes should be evaluated against observed albedo. The authors provide a very rough comparison with ranges of albedo measurements on p.7 line 8, but I think a spatial evaluation against observations from a space-borne sensor like MODIS would be much more useful. Although it is interesting and important to know how the different albedo schemes affect simulated SMB and ice thickness, it is also critical, in my opinion, to know how the albedo scheme performs in terms of albedo.**

In response to this issue, and also to issues raised by the other reviewers, we added a comparison with albedo as simulated by RACMO2. We chose to use albedo from a regional climate model instead of observed (MODIS) albedo, because now we evaluate albedo likewise other parameters (snowfall, melt, SMB), that were also compared to the output from RACMO2. As such, we included additional figures (Figure 2), containing the difference with RACMO2 JJA albedo over Greenland.

For more details about the comparison of albedo with RACMO2 albedo, see response to reviewer #1.

**(2) Related to (1): The determination of optimal albedo parameters seems flawed because, although these parameters produce the lowest RMSD in ice thickness (figure 8), they clearly do not produce the best simulation of SMB (in comparison with RACMO), as shown in Table 2. Thus, I am concerned that these albedo parameters produce the best simulation of ice thickness for the wrong reasons.**

Yes, we agree. It might be the case that the choice for this albedo scheme compensates for biases associates to other processes that are not (correctly) simulated. As such, we added the following in the Discussion section (P12, L22-25):

However, we do not claim that our proposed albedo scheme is physically

superior to other schemes. Due to error compensation, the best parameters as found here make up for other unknown model flaws, which might lead to errors in future projections. Nevertheless, based on the ice sheet simulations, the "Utr-8" albedo parameterization seems the best parameter setting within the set of albedo schemes to be used in EC-Earth simulations with an interactive ice sheet component.

**(3) The authors note (p. 6,21) that "Since the simulated climate is freely evolving after initialization, a direct comparison with observed time series is not meaningful." Could land-only simulations be performed, i.e., with fixed atmosphere conditions? Such simulations would enable a comparison of the albedo schemes that is much less affected by weather/climate variability, and would enable a meaningful comparison with observed time series.**

We perform atmosphere-only experiments. Unfortunately it is not feasible to perform land-only simulations within EC-Earth.

**Minor comments:**

**p.3,1: "its interaction" -> "their interactions"**

We modified this line as follows, such that it is clearer (P3, L4):

To better understand changes in the mass of the ice sheet and in its interaction with the climate system ...

**p.4,33: What is the justification for choosing -2K as the threshold for applying "wet" snow conditions? Are the results at all sensitive to this temperature choice?**

We follow the choice of the original albedo scheme by Dutra et al (2010). The motivation for this is that it "accounts for the subgrid-scale variability of the snowpack properties for typical climate and NWP resolutions" (Dutra et al., 2010).

**Sections 2.2.1 and 2.2.2: The Stockholm and Copenhagen schemes seem unrealistic because they, respectively, neglect dry snow aging and prescribe sharp, instantaneous changes to albedo under wet conditions, e.g., when snow temperature exceeds -2K. I believe these experiments are still useful because they represent bounds, but I suggest acknowledging a bit more clearly that they should not be viewed as realistic albedo schemes and are instead used for boundary-case sensitivity studies. I would also note that abrupt binary changes in physical parameters are often undesirable in climate models because they can lead to instabilities.**

The Stockholm scheme can be regarded as a first step in allowing the albedo to adjust over the ice sheet. The Copenhagen scheme makes a different choice, by using an instantaneous reaction to wet conditions. The rationale behind this is

written in section 2.2.2: "that the presence of water can have an instantaneous effect". As such, we feel that there is room for different views upon which scheme is most desirable in a climate model. Thus, we refrain from speaking out our preference, at this point. In the end, we do note that with our goal of simulating a Greenland Ice Sheet with a size that reasonably resembles the present-day state, we prefer the "Utr-8" scheme.

**p.6 31: "JJA mean pattern" - pattern of what? Please specify.**

After revising the manuscript on the subject of JJA albedo values , in relation to the comparison with RACMO2, this part of the text is rewritten. When we refer to Figure 2, we now more clearly state that 23-yr mean JJA albedo values are plotted.

**p.7,25: Why use the skin temperature to determine conditions for snow melting to occur? Is snow temperature not prognosed in the model?**

Yes, snow temperature is also prognosed in the model, but as the surface energy balance is made for the surface layer, we prefer to use skin temperature instead of the temperature of a snow layer of variable thickness.

**p.8,24: "A side effect of the use of a time-evolving albedo scheme is an increase of interannual variability in melt..." - Yes, but this should be a more realistic feature! (I mention this because "side effect" often connotes an adverse unintended consequence).**

True, we changed this to: An additional effect …

**p.9,28: "tje"**

Corrected

**p.10,16: How are the (relatively short) SMB fields applied in the 25-ky runs? Are the multi-year SMB timeseries (only 25 years in length, I think) simply repeated? Or are they averaged and repeated annually?**

They are averaged over the simulated period (1990-2012), and repeated annually. The paragraph describing how we deal with the SMB forcing for the ISM is now more extensively rewritten (P11, L14-20):

Several 25-ky runs are carried out using the 23-year mean SMB forcing fields resulting from the different choices in albedo parameters. To take into account the influence of topography changes on surface temperature, a uniform lapse rate of -7.4 K km$^{-1}$ is applied. The influence of topography changes on SMB (height-mass balance effect) is parameterized by calculating a new SMB forcing field each time step using SMB gradients (Helsen et al., 2012). These gradients are computed for each grid point, from a correlation between surface elevation and mean SMB values in an area with a radius of 150 km. This allows the ice sheet to advance outside the initial ice sheet mask, but also can lead to substantial retreat when ice sheet thinning occurs (Helsen et al., 2012).

**p.10,17: "A uniform lapse rate of -7.4 K km-1 is applied." - Why is this lapse rate used? It seems steeper than values commonly used in other downscaling studies.**

We use this lapse rate in agreement with earlier work (Helsen et al., 2012; 2013). We estimated this lapse rate from T2m values from RACMO2 data over Greenland.

**p.11,1: "downscaling the locally strongly negative SMB values are" - Please fix for clarity.**

We have rewritten these lines (P11, L34 – P12, L2):

Local ablation zones are not everywhere well captured by the SMB products, when the resolution of the climate model is not fine enough. In these cases, applying the downscaling method does not make much of a difference, as the resulting SMB gradients are not steep enough. This underestimation of ablation area results in ice sheet advance.

**p.13,2-4: "The height - mass balance..." - This sentence (particularly the reference to "given back to the climate model") is somewhat unclear to me.**

This statement is more an outlook of the possibilities when we apply this albedo parameterization in a coupled EC-Earth – ISM simulation.

**Figure 6: I suggest showing RMSD statistics for the SMB in this figure, as shown in figure 8 for ice thickness.**

RMSD values have been added for all figures that contain a comparison with RACMO2 fields, and references in the text are added if necessary.

---

## Referee Report (RR1)

**On the importance of the albedo parameterization for the surface mass balance of the Greenland ice sheet in EC-Earth**

Michiel Helsen, Roderik van de Wal, Thomas Reerink, Richard Bintanja, Marianne Sloth Madsen, Shuting Yang, Qiang Li, and Qiong Zhang

**General Comments:**

The manuscript is improved over the previous version. The limitations of the albedo scheme and model simulations have been made clear. The study highlights many of the challenges involved and improvements that can be made in modeling ice sheet mass balance in a GCM. It indicates the importance of capturing both albedo and refreezing accurately for simulations of both SMB and total ice sheet mass balance.

I think the authors should try to make clear the importance of including schemes that are not currently included in the GCM, such as a representation of bare ice albedo and a refreezing scheme, the importance of accurately capturing the spatial distribution of SMB for input to an ISM, and the importance of capturing ice flow and ice conditions properly in the ISM. These are important results of the study that the modeling community should be aware of, and more important than the finding that one particular configuration of the albedo scheme works best with the current state of the model, when used to force an ISM simulation. In fact the "best" albedo configuration improves simulation with the ISM, but doesn't improve SMB as compared to RACMO2. Additional minor modifications to the text to further emphasize these points would improve the manuscript.

**Specific Comments:**
1. **Title:** Perhaps "surface mass balance" could be changed to "mass balance", since the authors do explore the link between albedo and overall ice volume through the ISM simulations.
2. **P. 1, Lines 5-9:** The abstract still doesn't mention any specific results of the study, for example, what impact changing the albedo scheme has on SMB, runoff, etc., the impact on overall ice volume when coupled with the ISM simulations, limitations of the albedo scheme, and potential improvements that can be made (for instance including refreezing) for more realistic GCM simulations.
3. **P. 3, Line 16:** Perhaps the authors could also mention here that another outcome of the work is the identification of modifications to the model that will likely improve future simulation of ice sheet mass balance in the GCM.
4. **P. 4, Line 4:** Clarify how the model was evaluated, e.g. "its performance was good in comparison with observational datasets".
5. **P. 4, Line 6:** Add "(T255)" after "truncated at wave number 255".
6. **P. 4, Line 31:** Change "but each group" to "but each of the groups mentioned above" for clarity.
7. **P. 5, Lines 5-6:** The discussion of this equation is still a bit confusing. Perhaps change "A continuous…snowfall flux (F=1 kg m$^{-2}$ h$^{-1}$)" to "When the snowfall rate (F) exceeds 10 kg m$^{-2}$ h$^{-1}$, snow albedo is fixed at $\alpha_{max}$. A continuous reset for smaller rates of snowfall is implemented to reduce the importance of small amounts of snowfall on surface albedo."
8. **P. 6, Line 14:** To make it clear that the value $\alpha_{max}$ doesn't vary within a single simulation, make it clear that a set of experiments is performed, e.g. "A set of simulations is performed in which the value of $\alpha_{max}$ is varied…"

9. **P. 7, Line 31:** Why use the period 1960-1989 for RACMO2, when the GCM simulation uses SSTs for the period 1990-2012?
10. **P. 7, Line 31 or 32:** I think the mention of the RACMO2 albedo scheme, mentioned later in the paragraph, should be added here before discussing the figures.
11. **P. 8, Line 8:** "albedo can occasionally drop" refers to temporal variability, but average values are being discussed here. Perhaps revise to read "The positive bias… indicates that in some locations, summertime albedo can drop to values lower than 0.5 (the minimum snow albedo used in this study)."
12. **P. 11, Line 18:** "linear regression" is probably more appropriate than "correlation" in this case.
13. **P. 11, Lines 23-24:** It is not necessarily true that the minimum albedo value is too low (although it probably is too low for snow). If the model accounts for refreezing, SMB may increase to the point that the ice sheet will be stable. Please note this here.
14. **P. 11, Lines 21-25:** It is interesting that in all the GCM simulations, SMB is lower than observed, but forcing with RACMO2 results in the most realistic simulation. This suggests that the spatial distribution of SMB is important to the evolution of the ice sheet, rather than the overall numbers. I think this is a key result that should be emphasized more in other parts of the manuscript.
15. **P. 12, Line 23:** Some of the flaws in the model are known, for example the lack of a refreezing scheme and the lack of a scheme for distinguishing between bare-ice vs. snow. Some of the flaws and limitations of the ice sheet model are also known. These should be mentioned here.
16. **P. 12, Lines 24-25:** Utr-8 produces the best representation of the current state of the ice sheet, when used to force the ice sheet model, but it produces an SMB that is too low relative to RACMO2. There are also substantial differences in terms of albedo. Please clarify that Utr-8 is chosen because of the agreement in terms of ice volume and area. The other discrepancies should be noted here. It should also be noted that adding more realistic schemes to both the GCM and ISM will also change which scheme produces the best results.
17. **P. 13, Line 31:** Perhaps change "solve the..." to "acount for…"
18. **P. 14, Lines 2-11:** As noted above, the results reveal more than the influence of the albedo scheme, but also reveal the importance of capturing the spatial distribution of SMB correctly, of accounting for refreezing in the model, and the need for improvements to ISM simulations. These points should also be noted.
19. **Figure 2, caption:** Specify the period for the RACMO average.

**Technical Corrections:**
1. **P. 6, Line 9:** Change "generally regarded weak" to "generally regarded as weak"
2. **P. 6, Line 25:** Change "from ERA-Interim" to "from the ERA-Interim"
3. **P. 7, Line 20:** Change "then the other schemes" to "than the other schemes"
4. **P. 7, Line 23:** Add "the" before "accumulation area".
5. **P. 7, Line 32:** Change "on a 11 km" to "at an 11 km"
6. **P. 8, Line 11:** Add "the" before "SMB through…"
7. **P. 8, Line 28:** Fix "froml", and change "which is extensively" to "which has been extensively"
8. **P. 9, Line 7:** Change "spatial resolution" to "spatial resolutions".
9. **P. 10, Line 21:** Change "as are associated changes in SMB" to "producing associated errors in SMB".

---

## Author Response (AR2)

**Response to reviewer #3**

*General Comments:*

*The manuscript is improved over the previous version. The limitations of the albedo scheme and model simulations have been made clear. The study highlights many of the challenges involved and improvements that can be made in modeling ice sheet mass balance in a GCM. It indicates the importance of capturing both albedo and refreezing accurately for simulations of both SMB and total ice sheet mass balance.*

*I think the authors should try to make clear the importance of including schemes that are not currently included in the GCM, such as a representation of bare ice albedo and a refreezing scheme, the importance of accurately capturing the spatial distribution of SMB for input to an ISM, and the importance of capturing ice flow and ice conditions properly in the ISM. These are important results of the study that the modeling community should be aware of, and more important than the finding that one particular configuration of the albedo scheme works best with the current state of the model, when used to force an ISM simulation. In fact the "best" albedo configuration improves simulation with the ISM, but doesn't improve SMB as compared to RACMO2. Additional minor modifications to the text to further emphasize these points would improve the manuscript.*

We thank the reviewer for his suggestions, and we agree fully with his opinion on this point. We have added and emphasized the above-mentioned points on different places in the manuscript:

*Abstract (p. 1, l. 6-9):*
*By allowing ice sheet albedo to vary as a function of wet and dry conditions, the spatial distribution of albedo and melt rate improves. Nevertheless, the spatial distribution of SMB in EC-Earth is not significantly improved. As a reason for this, we identify omissions in the current snow albedo scheme, such as separate treatment of snow and ice and the effect of refreezing.*

*Introduction (p.3, l. 22-24):*
*Our experiments also identify which modifications are necessary for further improvements of ice sheet mass balance within GCMs. Such improvements regarding the description of ice sheets in GCMs is vital for a better understanding of changes in ice sheets in the past, present and future.*

*Ice sheet simulations (p.12, l. 4-6):*
*The ISM run forced by the RACMO2 climatology results in an ice volume and area most comparable to the present-day state (Howat et al., 2014), in spite of the fact that several EC-Earth schemes result in lower ice-sheet integrated SMB (Table 2). This*

*points out that the spatial distribution of SMB is important to the evolution of the ice sheet, rather than the overall numbers.*

*Discussion (p. 13, l. 34-35):*
*Moreover, accurately describing the effect of refreezing will improve the spatial distribution of SMB, which is of large importance for interactive climate – ice sheet model simulations.*

*Conclusion (p. 14, l. 25-31):*
*However, based on the different results obtained with a climate forcing from a RCM, our results emphasize the importance of capturing the spatial distribution of the SMB, rather than the ice-sheet integrated number. We note that the physics of the albedo scheme can still be greatly improved with the inclusion of a multi-layer snow model component in the land surface component of EC-Earth, to better account for refreezing of percolating meltwater in snow, and to distinguish between bare ice and snow. Hence, further improvements of the snow scheme are crucial for the development of earth system models including an interactive ice sheet component.*

**Specific Comments:**
**1. Title: Perhaps "surface mass balance" could be changed to "mass balance", since the authors do explore the link between albedo and overall ice volume through the ISM simulations.**

Done, new title:
On the importance of the albedo parameterization for the mass balance of the Greenland ice sheet in EC-Earth

**2. P. 1, Lines 5-9: The abstract still doesn't mention any specific results of the study, for example, what impact changing the albedo scheme has on SMB, runoff, etc., the impact on overall ice volume when coupled with the ISM simulations, limitations of the albedo scheme, and potential improvements that can be made (for instance including refreezing) for more realistic GCM simulations.**

Done, the following lines are added to the abstract:
*Abstract (p. 1, l. 6-9):*
*By allowing ice sheet albedo to vary as a function of wet and dry conditions, the spatial distribution of albedo and melt rate improves. Nevertheless, the spatial distribution of SMB in EC-Earth is not significantly improved. As a reason for this, we identify omissions in the current snow albedo scheme, such as separate treatment of snow and ice and the effect of refreezing.*

**3. P. 3, Line 16: Perhaps the authors could also mention here that another outcome of the work is the identification of modifications to the model that will likely improve future simulation of ice sheet mass balance in**

**the GCM.**
We agree on this. We added the following to the introduction:
*Introduction (p.3, l. 22-24):*
*Our experiments also identify which modifications are necessary for further improvements of ice sheet mass balance within GCMs. Such improvements regarding the description of ice sheets in GCMs is vital for a better understanding of changes in ice sheets in the past, present and future.*

**4. P. 4, Line 4: Clarify how the model was evaluated, e.g. "its performance was good in comparison with observational datasets".**
Done, we added the following (p. 4, l. 11-13):
*EC-Earth (version 2.3) participated in CMIP5 (Taylor et al., 2012). The model was evaluated against observations, reanalysis data and other coupled atmosphere-ocean-sea ice models, and its performance was good, in terms of the mean state, spatial patterns, seasonal cycle and variability of present-day climate (Hazeleger et al., 2010, 2012).*

**5. P. 4, Line 6: Add "(T255)" after "truncated at wave number 255".**
Done

**6. P. 4, Line 31: Change "but each group" to "but each of the groups mentioned above" for clarity.**
Done

**7. P. 5, Lines 5-6: The discussion of this equation is still a bit confusing. Perhaps change "A continuous...snowfall flux ($F=1$ kg m$^{-2}$ h$^{-1}$)" to "When the snowfall rate (F) exceeds 10 kg m$^{-2}$ h$^{-1}$, snow albedo is fixed at $\alpha_{max}$. A continuous reset for smaller rates of snowfall is implemented to reduce the importance of small amounts of snowfall on surface albedo."**
Done

**8. P. 6, Line 14: To make it clear that the value $\alpha_{max}$ doesn't vary within a single simulation, make it clear that a set of experiments is performed, e.g. "A set of simulations is performed in which the value of $\alpha_{max}$ is varied..."**
Done.

**9. P. 7, Line 31: Why use the period 1960-1989 for RACMO2, when the GCM simulation uses SSTs for the period 1990-2012?**
This is perhaps not the best choice, in hindsight. We used this part of the RACMO2 dataset since we assume that the GrIS is in equilibrium with the climate forcing in this period.

**10. P. 7, Line 31 or 32: I think the mention of the RACMO2 albedo scheme, mentioned later in the paragraph, should be added here before discussing the figures.**

Done

**11. P. 8, Line 8: "albedo can occasionally drop" refers to temporal variability, but average values are being discussed here. Perhaps revise to read "The positive bias... indicates that in some locations, summertime albedo can drop to values lower than 0.5 (the minimum snow albedo used in this study)."**

Done

**12. P. 11, Line 18: "linear regression" is probably more appropriate than "correlation" in this case.**

Done

**13. P. 11, Lines 23-24: It is not necessarily true that the minimum albedo value is too low (although it probably is too low for snow). If the model accounts for refreezing, SMB may increase to the point that the ice sheet will be stable. Please note this here.**

Done (p. 11, l. 32 – p. 12, l. 3):

*This is expected, as it is the only climatology with an initial negative SMB (Table 2), which suggests that the $\alpha_{min}$ value of 0.45 of albedo scheme "Utr-9" is too low. However, this result might be different if the model would account for refreezing, which would lead to a higher SMB, perhaps up to the point that the ice sheet will be stable. Hence, our results are strongly determined by the characteristics of our snow scheme.*

**14. P. 11, Lines 21-25: It is interesting that in all the GCM simulations, SMB is lower than observed, but forcing with RACMO2 results in the most realistic simulation. This suggests that the spatial distribution of SMB is important to the evolution of the ice sheet, rather than the overall numbers. I think this is a key result that should be emphasized more in other parts of the manuscript.**

Done, see the different additional lines as indicated on page 1 of this document.

**15. P. 12, Line 23: Some of the flaws in the model are known, for example the lack of a refreezing scheme and the lack of a scheme for distinguishing between bare-ice vs. snow. Some of the flaws and limitations of the ice sheet model are also known. These should be mentioned here.**

Done.

**16. P. 12, Lines 24-25: Utr-8 produces the best representation of the current state of the ice sheet, when used to force the ice sheet model,**

**but it produces an SMB that is too low relative to RACMO2. There are also substantial differences in terms of albedo. Please clarify that Utr-8 is chosen because of the agreement in terms of ice volume and area. The other discrepancies should be noted here. It should also be noted that adding more realistic schemes to both the GCM and ISM will also change which scheme produces the best results.**

Done (p. 13, l. 7-10):

*Nevertheless, based on the agreement in ice sheet area and volume, the "Utr-8" albedo parameterization seems the best parameter setting within the set of albedo schemes to be used in EC-Earth simulations with the current snow scheme and an interactive ice sheet component. Adding a more sophisticated snow scheme will likely change optimal choices for albedo parameters.*

**17.  P. 13, Line 31: Perhaps change "solve the..." to "acount for..."**
Done.

**18.  P. 14, Lines 2-11: As noted above, the results reveal more than the influence of the albedo  scheme, but also reveal the importance of capturing the spatial distribution of SMB correctly, of accounting for refreezing in the model, and the need for improvements to ISM simulations. These points should also be noted.**

Done, the conclusion now is as follows (p. 14, l. 18-31):

*We have extended the albedo parameterization over the GrIS in the earth system model EC-Earth, to replace the constant value of 0.80 over perennial snow in EC-Earth. We applied different exponentially-decaying functions to account for the slow and fast response of $\alpha_{sn}$ in dry and wet conditions, respectively. Our results show that small adjustments to the albedo scheme significantly influence the SMB over Greenland. This in turn affects the ice sheet response, implying consequences for coupled ice sheet – climate simulations in an earth system model framework. The height - mass balance effect that we parameterized here using a relation of SMB with surface elevation will be more accurately solved when ice sheet elevation and extent are given back to the climate model. Based on the ice sheet simulations, the "Utr-8" albedo parameterization seems the most suitable albedo scheme to be used in EC-Earth simulations with an interactive ice sheet component. However, based on the different results obtained with a climate forcing from a RCM, our results emphasize the importance of capturing the spatial distribution of the SMB, rather than the ice-sheet integrated number. We note that the physics of the albedo scheme can still be greatly improved with the inclusion of a multi-layer snow model component in the land surface component of EC-Earth, to better account for refreezing of percolating meltwater in snow, and to distinguish between bare ice and snow. Hence, further improvements of the snow scheme are crucial for the development of earth system models including an interactive ice sheet component.*

**19.    Figure 2, caption: Specify the period for the RACMO average.**

Done

**Technical Corrections:**

**1.P. 6, Line 9: Change "generally regarded weak" to "generally regarded as weak"**

Done

**2.P. 6, Line 25: Change "from ERA-Interim" to "from the ERA-Interim"**

Done

**3.P. 7, Line 20: Change "then the other schemes" to "than the other schemes"**

Done

**4.P. 7, Line 23: Add "the" before "accumulation area".**

Done

**5.P.7,Line32: Change"ona11km"to"atan11km"**

Done

**6.P. 8, Line 11: Add "the" before "SMB through..."**

Done

**7.P. 8, Line 28: Fix "froml", and change "which is extensively" to "which has been extensively"**

Done

**8.P. 9, Line 7: Change "spatial resolution" to "spatial resolutions".**

Done

**9.P. 10, Line 21: Change "as are associated changes in SMB" to "producing associated errors in  SMB".**

Done